# Suppressed prefrontal neuronal firing variability and impaired social representation in IRSp53-mutant mice

**Woohyun Kim[1], Jae Jin Shin[2], Yu Jin Jeong[3], Kyungdeok Kim[1,2], Jung Won Bae[1], Young Woo Noh[1], Seungjoon Lee[1], Woochul Choi[4], Se-Bum Paik[4], Min Whan Jung[1,2]\*, Eunee Lee[3]\*, Eunjoon Kim[1,2]\***

[1]Department of Biological Sciences, KAIST, Daejeon, Republic of Korea; [2]Center for Synaptic Brain Dysfunctions, Institute for Basic Science (IBS), Daejeon, Republic of Korea; [3]Department of Anatomy, College of Medicine, Yonsei University, Seoul, Republic of Korea; [4]Department of Bio and Brain Engineering, KAIST, Daejeon, Republic of Korea

**Abstract** Social deficit is a major feature of neuropsychiatric disorders, including autism spectrum disorders, schizophrenia, and attention-deficit/hyperactivity disorder, but its neural mechanisms remain unclear. Here, we examined neuronal discharge characteristics in the medial prefrontal cortex (mPFC) of IRSp53/Baiap2-mutant mice, which show social deficits, during social approach. We found a decrease in the proportion of IRSp53-mutant excitatory mPFC neurons encoding social information, but not that encoding non-social information. In addition, the firing activity of IRSp53-mutant neurons was less differential between social and non-social targets. IRSp53-mutant excitatory mPFC neurons displayed an increase in baseline neuronal firing, but decreases in the variability and dynamic range of firing as well as burst firing during social and non-social target approaches compared to wild-type controls. Treatment of memantine, an NMDA receptor antagonist that rescues social deficit in IRSp53-mutant mice, alleviates the reduced burst firing of IRSp53-mutant pyramidal mPFC neurons. These results suggest that suppressed neuronal activity dynamics and burst firing may underlie impaired cortical encoding of social information and social behaviors in IRSp53-mutant mice.

\*For correspondence:
mwjung@kaist.ac.kr (MWhanJ);
obsee93@gmail.com (EL);
kime@kaist.ac.kr (EK)

## Editor's evaluation

This study by Kim et al. is of interest to neuroscientists studying neocortical neural activity, as related to social behavior and in mouse models of neuropsychiatric disorders. These results provide new data on how the loss of the postsynaptic scaffolding and adaptor protein IRSp53 impacts prefrontal cortex activity and social interaction in mice. The authors propose the interesting idea that suppressed neuronal activity dynamics and burst firing may contribute to the impaired cortical encoding of social information and social behaviors in IRSp53-mutant mice.

## Introduction

Social dysfunction is a key feature of various neuropsychiatric disorders, including autism spectrum disorders (ASD), schizophrenia, and attention-deficit/hyperactivity disorders (ADHD). Among the various brain regions involved in social regulation, the medial prefrontal cortex (mPFC) plays critical roles in integrative and higher cognitive brain functions (*Yan and Rein, 2022*; *Yizhar and Levy, 2021*). Previous studies identified a number of mechanisms associated with dysfunctions under a

social context. Examples include an imbalance of neuronal excitation/inhibition (*Selimbeyoglu et al., 2017*; *Yizhar et al., 2011*) (reviewed in *Lee et al., 2017*; *Nelson and Valakh, 2015*; *Sohal and Ruben-stein, 2019*), impaired cortical social representation (*Lee et al., 2021a*; *Lee et al., 2021b*; *Lee et al., 2016*; *Levy et al., 2019*; *Miura et al., 2020*), and the disruption of local oscillations (*Cao et al., 2018*; *Yizhar et al., 2011*). Given that social behaviors represent outcomes of complex interactions among multiple underlying neural processes, further mechanistic explorations are needed to be investigated in the context of additional genes and various psychiatric disorders.

Insulin receptor substrate protein 53 kDa (IRSp53) encoded by the *BAIAP2* gene is a postsynaptic scaffolding and adaptor protein at excitatory synapses that interacts with other key components of the postsynaptic density such as PSD-95 (*Choi et al., 2005*; *Soltau et al., 2004*). IRSp53 has also been implicated in ASD (*Celestino-Soper et al., 2011*; *Levy et al., 2011*; *Toma et al., 2011*; *Wu et al., 2020*), schizophrenia (*Fromer et al., 2014*; *Genovese et al., 2016*; *Johnson et al., 2016*; *Purcell et al., 2014*) and ADHD (*Bonvicini et al., 2016*; *Liu et al., 2013*; *Ribasés et al., 2009*). Functionally, IRSp53 regulates actin filament dynamics at excitatory synapses and dendritic spines (*Kang et al., 2016*; *Scita et al., 2008*).

IRSp53 deficiency in mice leads to excitatory synaptic deficits and various behavioral deficits, including hyperactivity, cognitive impairments, and social deficits (*Bobsin and Kreienkamp, 2016*; *Chung et al., 2015*; *Kim et al., 2009*; *Kim et al., 2020b*; *Sawallisch et al., 2009*). IRSp53 knockout (KO) mice have fewer dendritic spines and enhanced NMDA receptor (NMDAR) function; they show impaired social behavior that is rescued by pharmacological NMDAR suppression (*Chung et al., 2015*; *Kim et al., 2009*). Importantly, mPFC neurons in IRSp53-KO mice show reduced neuronal firing under urethane-anesthesia, which is acutely normalized by pharmacological NMDAR suppression (*Chung et al., 2015*). However, it remained unknown whether and how the social behavioral deficits are associated with altered mPFC neural activity in waking-state animals engaged in social interaction.

To study the neural abnormalities of the mPFC associated with social dysfunction in IRSp53-KO mice, we herein performed single-unit recordings in freely moving mice engaged in social interactions in a linear social-interaction chamber (*Lee et al., 2016*). We found that excitatory neurons in the mPFC of IRSp53-KO mice display narrower dynamic ranges of firing rate and reduced burst firing. As a consequence, they showed lower discrimination between social and object targets compared to those of wild-type (WT) control. Our results uncover a novel social coding deficit associated with IRSp53-KO.

## Results
### Social impairments in IRSp53-KO mice in the linear-chamber social-interaction test

To compare neuronal activities in the mPFC of WT and IRSp53-KO mice during social interaction, we performed single-unit recordings in mice engaged in social interaction in a linear-chamber social-interaction apparatus (*Figure 1A*). The chamber, a long corridor connected with two side chambers with targets, was designed to measure neuronal activity during social interaction (*Lee et al., 2016*). A subject mouse was first placed in a separate circular rest box for 5 min for recording of resting neural activity. The mouse was then placed into the linear social-interaction chamber and allowed to explore the chamber with both side chambers being empty (empty-empty/E-E session) for 10 min. This was followed by a session in which one of the side chambers contained a novel social target (S; a conspecific male mouse) and the other contained a novel inanimate object (O) (first S-O session), and another session where the S and O were switched (second S-O session), which was included to control for side (or location)-specific as opposed to target-specific neural activity. The positions of mice in the linear chamber during experiments were determined using the DeepLabCut program (*Lauer et al., 2022*; *Mathis et al., 2018*), which automatically marked the subject mouse's nose, ears, body center, and tail base as well as the social target's nose and body center (*Figure 1B*). Sniffing time was defined as the time when the mouse's nose was within a distance of 3 cm from the front face of the target chamber. In-zone time was defined as the time when the body center (midpoint between the nose and tail base) fell in the area within 9 cm from the front face of the target chamber.

The single-unit activity was recorded with tetrodes from the prelimbic (PrL), infralimbic (IL), and cingulate cortex (Cg1) regions. Eight tetrodes, four tetrodes in each hemisphere, were implanted into the mPFC and lowered after each round of recording experiment to record neurons at different

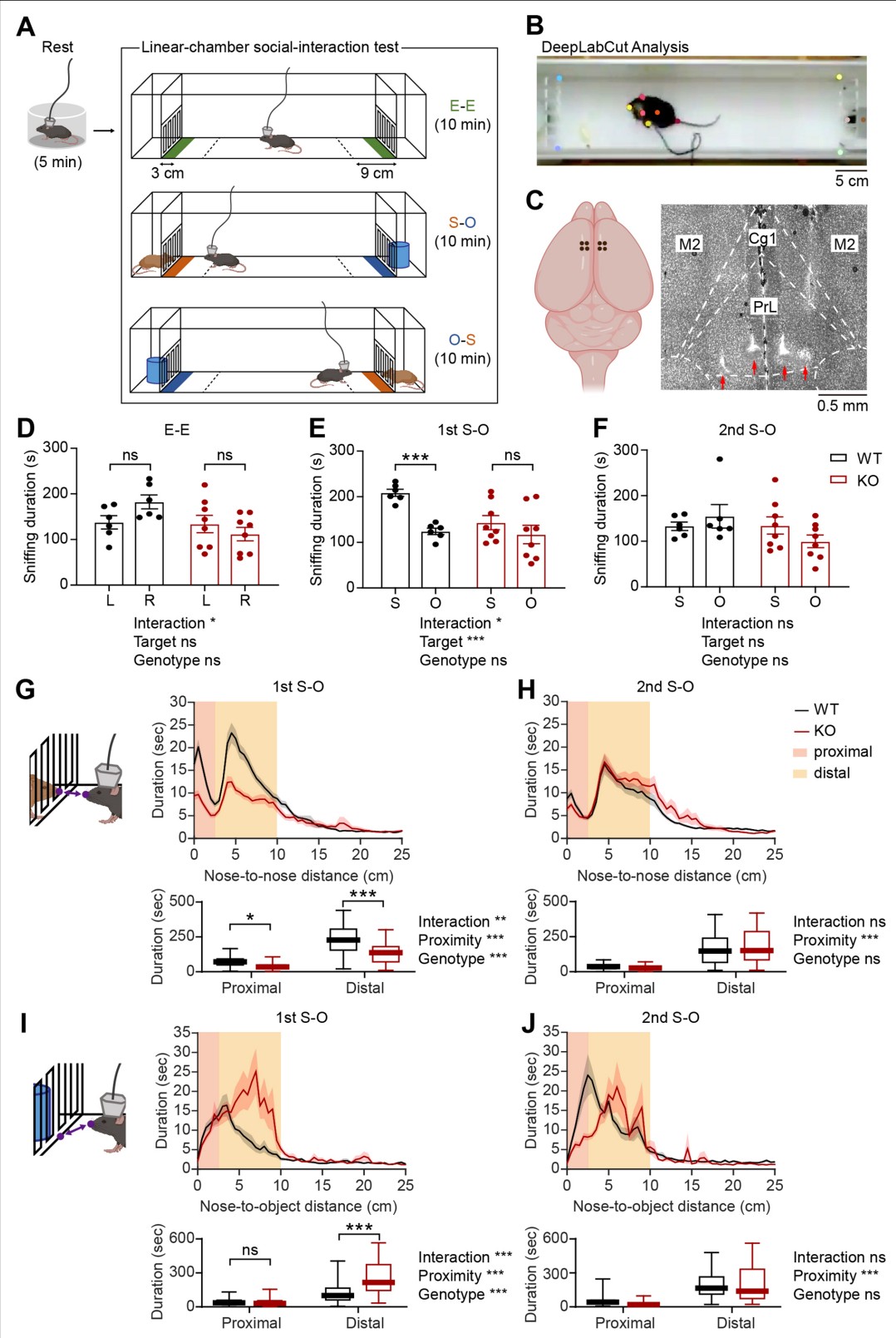

**Figure 1.** Social impairments in IRSp53-KO mice in the linear-chamber social-interaction test. (**A**) Schematic diagram of the linear-chamber social-interaction test used to measure social approach towards a novel conspecific mouse (S, social) versus a novel non-social target (O, object). A tetrode-implanted mouse was first placed in the rest box for 5 min and moved to the linear social-interaction chamber to perform the following three sessions: empty-empty (**E–E**) session, social-object (first S-O) session, and object-social (second S-O) session. The in-zone area, falling within 9 cm from the front

*Figure 1 continued on next page*

*Figure 1 continued*

face of the chambers, is indicated by a dashed line. The sniffing zone, falling within 3 cm from the front face of the chambers, is indicated by green, orange, and blue colors. (**B**) An example video frame of subject mouse and social target mouse body parts automatically tracked by the DeepLabCut program. (**C**) Schematic (left) and a representative coronal brain section (right) showing the locations of the implanted tetrodes. PrL, prelimbic cortex; IL, infralimbic cortex; Cg1, cingulate cortex, area 1; M2, secondary motor cortex. (**D–F**) Mean (± standard error of mean/SEM) sniffing duration for left (**L**) vs. right (**R**) empty targets during the E-E session (**D**) and the social (**S**) vs. object (**O**) targets during the first S-O (**E**) and second S-O (**F**) sessions. (n=6 mice [WT], 8 mice [IRSp53-KO], *$P<0.05$, ***$P<0.001$, ns, not significant, two-way repeated-measures (RM)-ANOVA with Sidak's multiple comparisons test). (**G and H**) Mean (± SEM) duration of the interaction between subject and social target mice as a function of the distance between the noses of the two mice during the first (**G**) and second (**H**) S-O sessions (top). Cumulative duration of proximal and distal social interactions (bottom). (n=47 experiments from 6 mice [WT], 58, 8 [IRSp53-KO], *$p<0.05$, **$p<0.01$, ***$p<0.001$, ns, not significant, two-way RM-ANOVA with Sidak's multiple comparison test). (**I and J**) Mean (± SEM) duration of the interaction between subject mice and object target as a function of the distance between the subject mouse's nose and the center of the object chamber face during the first (**I**) and second (**J**) S-O session (top). Cumulative duration of proximal and distal object interactions (bottom). (n=47 experiments from 6 mice [WT], 58, 8 [IRSp53-KO], ***$p<0.001$, ns, not significant, two-way RM-ANOVA with Sidak's multiple comparison test). See *Supplementary file 2* for statistics. Numerical data used to generate the figure are available in the *Figure 1—source data 1*.

The online version of this article includes the following source data and figure supplement(s) for figure 1:

**Source data 1.** Source file for mouse behavior data in *Figure 1*.

**Figure supplement 1.** Locations of implanted tetrodes in the mPFC of WT and IRSp53-KO mice.

**Figure supplement 2.** Social impairments, as judged by in-zone durations, and unaltered mean duration of each visit or locomotor activity in IRSp53-KO mice in the linear social-interaction chamber.

**Figure supplement 2—source data 1.** Source file for mouse behavior data in *Figure 1—figure supplement 2*.

**Figure supplement 3.** Comparable composition of behavior in social and object target sniffing zones in WT and IRSp53-KO mice in the linear social-interaction chamber.

**Figure supplement 3—source data 1.** Source file for mouse behavior data in *Figure 1—figure supplement 3*.

depths. After the last recording, the locations of all tetrodes were assessed via histology, and data from those falling within the area of interest were used for analysis (*Figure 1C*, *Figure 1—figure supplement 1A*).

In the E-E session, WT and IRSp53-KO mice showed a preference for neither chamber, as assessed by sniffing and in-zone durations (*Figure 1D*, *Figure 1—figure supplement 2A*). In the first S-O session, IRSp53-KO mice spent a comparable amount of time exploring the social and object targets, whereas WT mice displayed a strong preference for the social target (*Figure 1E*, *Figure 1—figure supplement 2B*). In the second S-O session, WT mice no longer displayed social preference, likely because of social habituation (*Figure 1F*, *Figure 1—figure supplement 2C*).

While IRSp53-KO mice showed a decreased number of sniffing visits to the social conspecific mouse, their mean duration of each visit was comparable to that of the WT mice (*Figure 1—figure supplement 2D and E*). The latter suggests that the impaired social preference is less likely to be caused by genotypic differences in olfactory processing speeds. In fact, a previous report has shown that IRSp53-KO mice display normal olfactory function in the buried food-seeking test (*Chung et al., 2015*). Moreover, there was no genotype difference in the total distance traveled (*Figure 1—figure supplement 2F and G*). WT and IRSp53-KO mice displayed a decline in locomotor activity across successive sessions (E-E, first S-O, and second S-O) in each recording experiment (*Figure 1—figure supplement 2F*), but their overall locomotion remained comparable across the ten experiments (*Figure 1—figure supplement 2G*).

Before comparing the activity of excitatory mPFC neurons during target sniffing, whether the target-interacting behaviors are comparable between genotypes in terms of the proximity to targets was assessed. For social interaction, the distance between the noses of the subject and target mice was examined. Social interactions were divided into proximal and distal according to the nose-to-nose distance (proximal, <2.5 cm; distal, between 2.5 and 10 cm). Total durations of both proximal and distal social interactions were reduced in IRSp53-KO mice during the first, but not second, S-O session (*Figure 1G and H*). For object interaction, the distance between the subject mouse's nose and the center of the object chamber face was assessed. There was an increase in the duration of distal but not proximal object interaction in IRSp53-KO mice during the first S-O session, although there was no genotype difference in the second S-O session (*Figure 1I and J*). In addition, the relative amounts of time spent for proximal versus distal interactions did not vary across genotypes during social or object sniffing (*Figure 1—figure supplement 3A and B*).

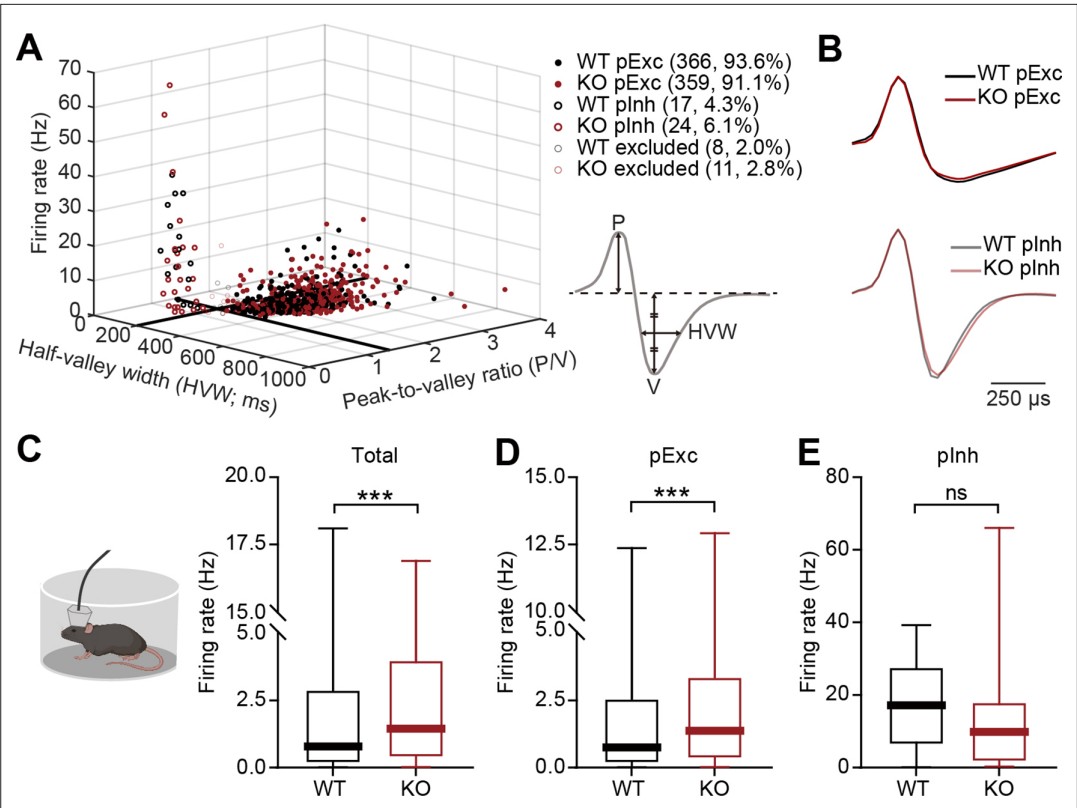

**Figure 2.** Increased resting firing rate in IRSp53-KO pExc mPFC neurons. (**A**) Classification of recorded neurons into putative excitatory (pExc) and putative inhibitory (pInh) neurons based on the half-valley width (200ms) and peak-to-valley ratio (1.4). P, peak; V, valley; HVW, half-valley width. (**B**) Average waveforms of WT and IRSp53-KO pExc (top) and pInh (bottom) neurons (n=366 [WT-pExc], 359 [KO-pExc], 17 [WT-pInh], 24 [KO-pInh]). The waveforms of each neuron were normalized by their peak values. (**C–E**) Firing rates of WT and IRSp53-KO total (**C**), pExc (**D**), and pInh (**E**) neurons in the mPFC during the 5-min rest period. (n=391 [WT-total], 394 [KO-total], 366 [WT-pExc], 359 [KO-pExc], 17 [WT-pInh], 24 [KO-pInh], ***p<0.001, ns, not significant, Mann-Whitney test). See **Supplementary file 2** for statistics. Numerical data used to generate the figure are available in the **Figure 2—source data 1**.

The online version of this article includes the following source data for figure 2:

**Source data 1.** Source file for resting firing rate data in **Figure 2**.

These behavioral results collectively indicate that IRSp53-KO mice display social impairments in the linear social-interaction chamber, similar to the social impairments previously determined using the three-chamber test, direct/dyadic social-interaction test, and ultrasonic vocalization test (**Chung et al., 2015**).

## Increased resting firing rate in IRSp53-KO pExc mPFC neurons

We next compared neuronal firing patterns in the mPFC of WT and IRSp53-KO mice during the above-mentioned linear-chamber social-interaction test. To this end, we first analyzed rest-period firing rates in awake and freely moving WT and IRSp53-KO mice. We segregated the neurons into putative excitatory (pExc) and putative inhibitory (pInh) neurons based on their half-valley width (pExc >200ms; pInh <200ms) and peak-to-valley ratio (pExc >1.4; pInh <1.4) (**Figure 2A and B**). The firing rate of total neurons at rest was higher in the mPFC of IRSp53-KO mice compared with WT mice (**Figure 2C**). However, only the IRSp53-KO pExc neurons, but not IRSp53-KO pInh neurons, showed a significant increase in firing rate (**Figure 2D and E**), suggesting that pExc neurons mainly contribute to the increase in the total firing rate. These results differ from those previously obtained from anesthetized IRSp53-KO mice (**Chung et al., 2015**), which exhibited decreases in total and pExc firing. This highlights the importance of measuring cortical neuronal activity in awake, behaving mice.

It should be noted that the majority of recorded neurons were pExc neurons (WT: 366 neurons, 93.6%, IRSp53-KO: 359 neurons, 91.1%), and that relatively few recordings were obtained from pInh neurons (WT: 17 neurons, 4.3%, IRSp53-KO: 24 neurons, 6.1%). Because IRSp53 is expressed primarily in the excitatory pyramidal (not inhibitory) neurons of the cortex (*Burette et al., 2014*), we hypothesized that the main effects of IRSp53 loss are seen in pExc neurons. Therefore, only pExc neurons were used for further analysis. Of all pExc neurons recorded, only those with a mean firing rate ≥0.5 Hz were included for further analysis in order to avoid low sampling errors arising from the inclusion of neurons with low firing rates (*Supplementary file 1*).

## Fewer social-responsive pExc mPFC neurons in IRSp53-KO mice

We compared target-dependent mPFC neuronal activity between IRSp53-KO and WT mice to test whether the social behavioral deficit found in IRSp53-KO mice is mirrored in mPFC neuronal activity. For this, we analyzed neuronal activity during the three linear chamber sessions (E-E, first S-O, and

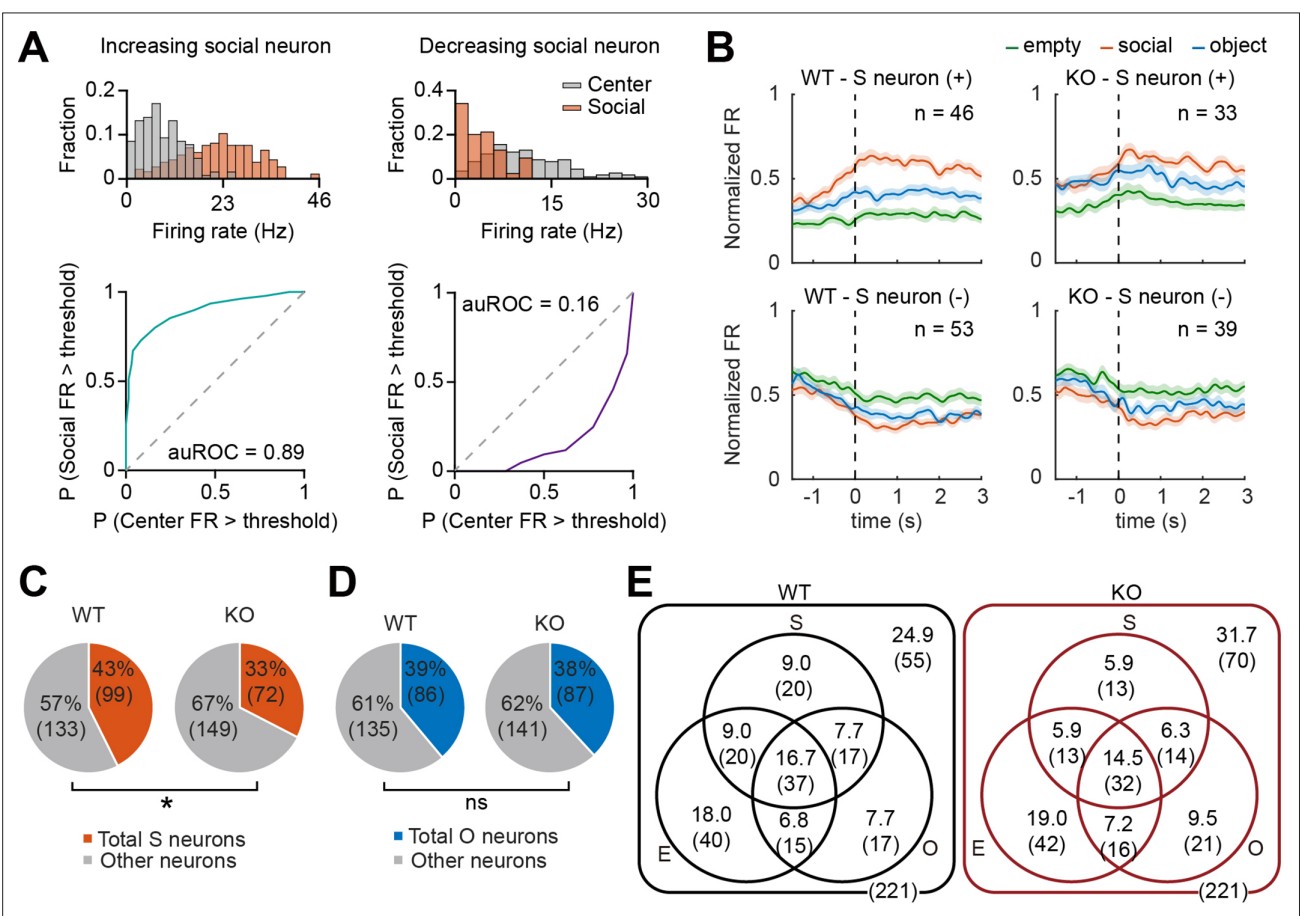

**Figure 3.** Fewer social pExc mPFC neurons in IRSp53-KO mice. (**A**) Distributions of instantaneous firing rates (FR) during social sniffing and center zone (top) and the receiver operating characteristic curves (ROCs; bottom) of increasing (left) and decreasing (right) social neuron examples. (**B**) Average spike density functions (SDFs) of firing rate responses to empty (green), social (orange), and object (blue) targets (aligned to the onset of sniffing) for all social (**S**) neurons. Social neurons are divided by genotype (WT left, IRSp53-KO right) and response direction (increasing (+) top, decreasing (-) bottom). Total numbers of neurons are indicated at the upper right corner of each SDF. Shading indicates ± SEM. (**C and D**) Proportions of total social (**C**) and object (**D**) neurons (both increasing and decreasing neurons) out of the total recorded neurons. (*p<0.05, ns, not significant, Fisher's exact test). (**E**) Venn diagram summary of empty (**E**), social (**S**), and object (**O**) neuronal proportions for WT (left) and IRSp53-KO (right) pExc neurons. Numbers indicate neuronal proportion % (n neurons). See *Supplementary file 2* for statistics.

The online version of this article includes the following source data and figure supplement(s) for figure 3:

**Figure supplement 1.** Normal number of empty pExc mPFC neurons in IRSp53-KO mice.

**Figure supplement 2.** Robust target-dependent responses of WT and IRSp53-KO pExc mPFC neurons.

**Figure supplement 2—source data 1.** Source file for target neuron data in *Figure 3—figure supplement 2*.

second S-O sessions) and determined empty, social, and object target-responsive neurons (termed empty, social, and object neurons hereafter) as those whose firing at the target sniffing zone differed significantly from that in the center zone (*Figure 3A*; see Methods).

In order to determine whether the classified social, object, and empty neurons increase or decrease their firing rates upon target sniffing, we generated the average spike density functions (SDFs) for WT and IRSp53-KO target neurons. We found both increasing and decreasing target neurons (i.e., those increasing and decreasing their firing rates upon target sniffing onset, respectively) in WT as well as IRSp53-KO mice (*Figure 3B*, *Figure 3—figure supplement 1A and B*).

A significantly lower proportion was classified as social neurons among IRSp53-KO pExc neurons compared to WT pExc neurons (*Figure 3C*). Meanwhile, the proportions of object and empty neurons were comparable between genotypes (*Figure 3D*, *Figure 3—figure supplement 1C*). The numbers of target neurons were summarized in Venn diagrams (*Figure 3E*).

Reduction in social/non-social neuronal proportion in mPFC is a phenomenon that is shared by several autism mouse models, such as Shank2-KO (*Lee et al., 2021a*) and Cntnap2-KO mice (*Levy et al., 2019*), and therefore, may be causally related to the social impairment seen in IRSp53-KO mice.

## Robust target-dependent responses of pExc mPFC neurons

We then tested whether the classified target neurons respond consistently to specific targets across sessions and across trials. The firing responses of WT and IRSp53-KO mPFC pExc neurons to social, object, and sidedness were consistent across the first and second S-O sessions, as indicated by positively correlated z-scores across the two sessions (*Figure 3—figure supplement 2A–D*). When we compared the response magnitude of neurons concerning the proximity to targets, both WT and IRSp53-KO neurons showed significantly higher response magnitudes during proximal than distal interactions with social and object targets (*Figure 3—figure supplement 2E and F*). In addition, social neurons in the mPFC do not respond to all social interactions but rather display 'trial-to-trial stochasticity' (*Liang et al., 2018*). This stochasticity was also present in the increasing and decreasing social neurons in the WT and IRSp53-KO mPFC (*Figure 3—figure supplement 2G*). Both WT and IRSp53-KO target neurons displayed trial consistencies that are higher than 50%, which corroborates that the responses of social and object neurons to the targets do not occur by chance. Although the proportion of mPFC social neurons was reduced in IRSp53-KO mice, their consistencies in firing rate responses to proximal target interactions were comparable to that in WT mice (*Figure 3—figure supplement 2H1*).

## Limited social versus object firing-rate discriminability in IRSp53-KO pExc mPFC neurons

To test whether the firing-rate discriminability between social and object targets may also be limited in IRSp53-KO neurons, we compared social- versus object-target in-zone firing rates. The slopes of linear regression and the degrees of dispersion (indicated by 95% confidence interval) for the left versus right (L vs. R) in-zone firing rates in the E-E session were comparable between genotypes (*Figure 4A*). In contrast, the slopes of the linear regression lines relating social and object (S vs. O) firing rates were biased towards the social firing rate in both genotypes in the first and second S-O sessions, indicating preferential responses to social to object targets (*Figure 4B*, *Figure 4—figure supplement 1A*). Additionally, the confidence interval tended to be narrower for IRSp53-KO pExc neurons, especially in the first S-O session, compared to WT pExc neurons for the S versus O firing rates, suggestive of limited discriminability (*Figure 4B*). Consistently, the absolute difference in firing rate for S versus O (an indication of discriminability) in the first S-O session was significantly lower in IRSp53-KO pExc neurons than WT pExc neurons (*Figure 4C*). Nevertheless, IRSp53-KO pExc neurons could still discriminate between social and object targets significantly better compared to the left versus right side discrimination in the E-E session (*Figure 4C*). This result suggests that IRSp53-KO mice have the ability to recognize social and object stimuli, albeit in a reduced degree than WT mice.

We next performed both single cell and ensemble decoding analyses using the support vector machine to further assess the ability of neurons to discriminate between social and object targets. The decoding performances of individual neurons and neuronal population were both comparable between genotypes for left versus right sidedness discriminability in the E-E session (*Figure 4D*, *Figure 4—figure supplement 1B*). In contrast, these decoding performances were significantly

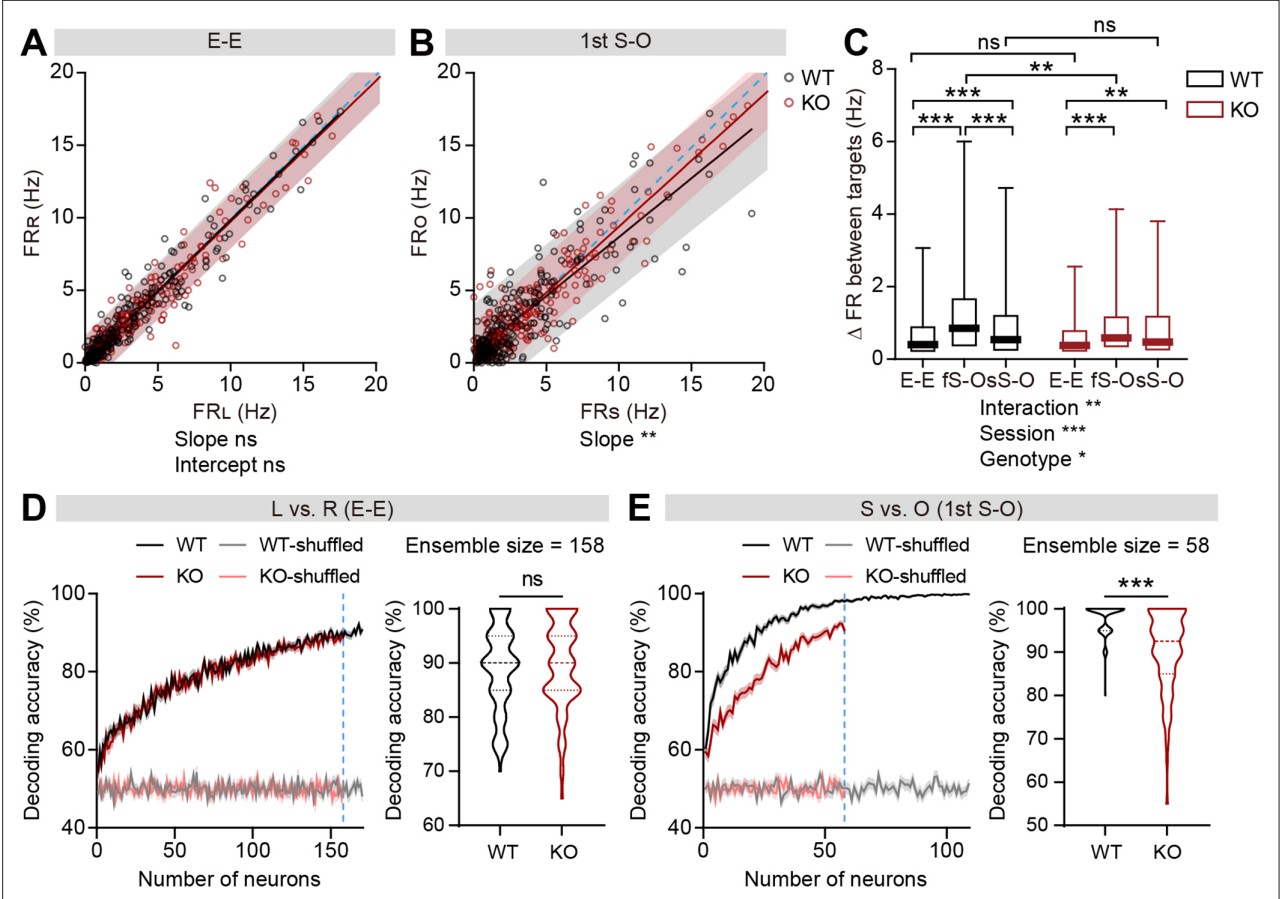

**Figure 4.** Limited discriminability between social and object targets in IRSp53-KO pExc mPFC neurons. (**A and B**) Scatterplot of left in-zone firing rate (FR$_L$) against right in-zone firing rate (FR$_R$) during the E-E session (**A**) and social in-zone firing rate (FR$_S$) against object in-zone firing rate (FR$_O$) during the first S-O session (**B**) for WT and IRSp53-KO pExc neurons. Solid lines indicate simple linear regressions for WT (black) and KO (red) neurons. Shaded areas indicate the 95% confidence intervals for the WT (black) and KO (red) firing rates. Blue dashed lines are 45 degree lines. (n=233 [WT-pExc] and 258 [KO-pExc]), **p<0.01, ns, not significant, simple linear regression with slope comparison test (see Methods). (**C**) Absolute changes in left versus right in-zone firing rates (E-E session) and social versus object in-zone firing rates (first (fS-O) and second (sS-O) S-O sessions) for WT and IRSp53-KO pExc neurons. (n=233 [WT-pExc] and 258 [KO-pExc], *p<0.05, **p<0.01, ***p<0.001, ns, not significant, two-way RM-ANOVA with Sidak's test). (**D and E**) Neural decoding of left versus right sidedness during the E-E session (**D**) and social versus object target during the first S-O session (**E**) as a function of ensemble size (left) and their decoding performance at maximum comparable ensemble size (indicated by blue dashed line) (right). Ensemble sizes vary due to the limitation of sniffing trials in some experiments (minimum trial number for the SVM decoding was set to 10 per target). Note that the decoding accuracies of WT (pink) and KO (grey) neurons remain similar to chance level (50%) across all tested ensemble sizes after target shuffling. (n=100 decoding trials for 170 and 109 pExc neurons in the E-E and first S-O sessions, respectively [WT] and 100, 158, 58 [KO], ***p<0.001, ns, not significant, Mann-Whitney test). See *Supplementary file 2* for statistics. Numerical data used to generate the figure are available in the *Figure 4—source data 1*.

The online version of this article includes the following source data and figure supplement(s) for figure 4:

**Source data 1.** Source file for firing-rate discriminability data in *Figure 4*.

**Figure supplement 1.** Comparable firing-rate discriminability between social and object targets in individual IRSp53-KO pExc mPFC neurons in the second S-O session.

**Figure supplement 1—source data 1.** Source file for firing-rate discriminability data in *Figure 4—figure supplement 1*.

**Figure supplement 2.** Increased firing-rate discriminability between social and object targets during proximal target interaction compared to distal target interaction.

**Figure supplement 2—source data 1.** Source file for firing-rate discriminability data in *Figure 4—figure supplement 2*.

lower in IRSp53-KO neurons for social versus object target discriminability in the first S-O session (*Figure 4E*, *Figure 4—figure supplement 1C*). In the second S-O session, although the individual neurons' decoding performances were comparable between genotypes, population decoding performances were significantly poorer in IRSp53-KO than WT neurons (*Figure 4—figure supplement 1D and E*).

Moreover, we examined whether the discriminability between social and object targets is affected by the proximity to the target. Although the discriminability was generally greater during proximal interaction compared to distal interaction, the discriminability in KO neurons was lower in both proximal and distal conditions compared to that in WT neurons in the first S-O session (*Figure 4—figure supplement 2A and B*). However, the discriminability was no longer different between genotypes during the distal social condition in the second S-O session (*Figure 4—figure supplement 2C and D*).

These results collectively suggest weakened social versus object discriminability in IRSp53-KO pExc mPFC neurons at both individual neuronal and population levels.

## Limited firing-rate range and variability of IRSp53-KO pExc mPFC neurons

The pExc mPFC neurons of IRSp53-KO mice showed significantly higher mean firing rates than those of WT mice during the initial 5-min rest period (*Figure 2D*), but comparable mean firing rates during the 30 min linear chamber test period (*Figure 5B*). However, we noticed in our preliminary analysis that temporal profiles of instantaneous firing rate (3 s time-bin advanced in 1 s steps) differ substantially between WT and IRSp53-KO neurons during the 30 min linear chamber test, as shown by representative examples in *Figure 5A*.

Further examinations of instantaneous firing rate revealed that the maximum instantaneous firing rate during the linear chamber test was significantly lower in IRSp53-KO pExc neurons than WT pExc neurons. In contrast, there was a trend for higher minimum instantaneous firing rates in IRSp53-KO pExc neurons than WT pExc neurons (p=0.0544; *Figure 5—figure supplement 1A and B*). Consequently, the dynamic range of firing rate (the difference between the maximum and minimum instantaneous firing rates) during the linear chamber test was significantly narrower for IRSp53-KO pExc neurons than WT pExc neurons (*Figure 5C*).

We next examined whether the limited firing-rate range is specific to social conditions. While the firing-rate range significantly increases from the E-E session to the first and second S-O sessions in WT neurons, these changes in firing-rate range were not observed in KO neurons (*Figure 5—figure supplement 1C*). Although the firing-rate range is comparable between genotypes in the resting state, it was significantly narrower in IRSp53-KO than WT mPFC neurons in both E-E (non-social) and S-O (social) conditions (*Figure 5—figure supplement 1D–G*). These results suggest that the limited firing-rate range in IRSp53-KO neurons is most salient during, but not restricted to, the social condition.

Another difference we noticed was that while WT neurons often remain silent and show abrupt increases in firing rate at specific time points, IRSp53-KO neurons tended to be active more chronically with their instantaneous firing rates fluctuating around the mean (*Figure 5A*). To test whether this is indeed the case, we examined the distribution of instantaneous firing rates of WT and IRSp53-KO neurons (normalized to the maximum firing rate). As expected, IRSp53-KO neurons had a significantly lower proportion of time-bins in the lowest firing rate (0–0.1) and instead higher proportions in mid-range firing rates (0.2–0.6) compared to the WT neurons (*Figure 5D*).

Given the reduced firing-rate range, we hypothesized that the firing rate variability of IRSp53-KO pExc neurons may also be decreased. We defined the firing rate variability of each neuron by the sigma value (1 standard deviation around the mean) of its instantaneous firing rates. We found that the instantaneous firing rates of IRSp53-KO neurons were indeed less variable, as indicated by a decrease in the sigma value (*Figure 5E*). In order to test if this decrease in variability in IRSp53-KO neurons is dependent upon the mean firing rate, the relationship between mean firing rates and sigma values was compared between genotypes (*Figure 5F*). As indicated by the significant decrease in the intercept—and comparable slope— of the linear regression, IRSp53-KO neurons were generally less variable in instantaneous firing activity regardless of their mean discharge rate. This decrease in sigma value was observed consistently across the analyses using variable sizes of time window for calculating instantaneous firing rate, ranging from 0.5 to 5 s (*Figure 5—figure supplement 1H*).

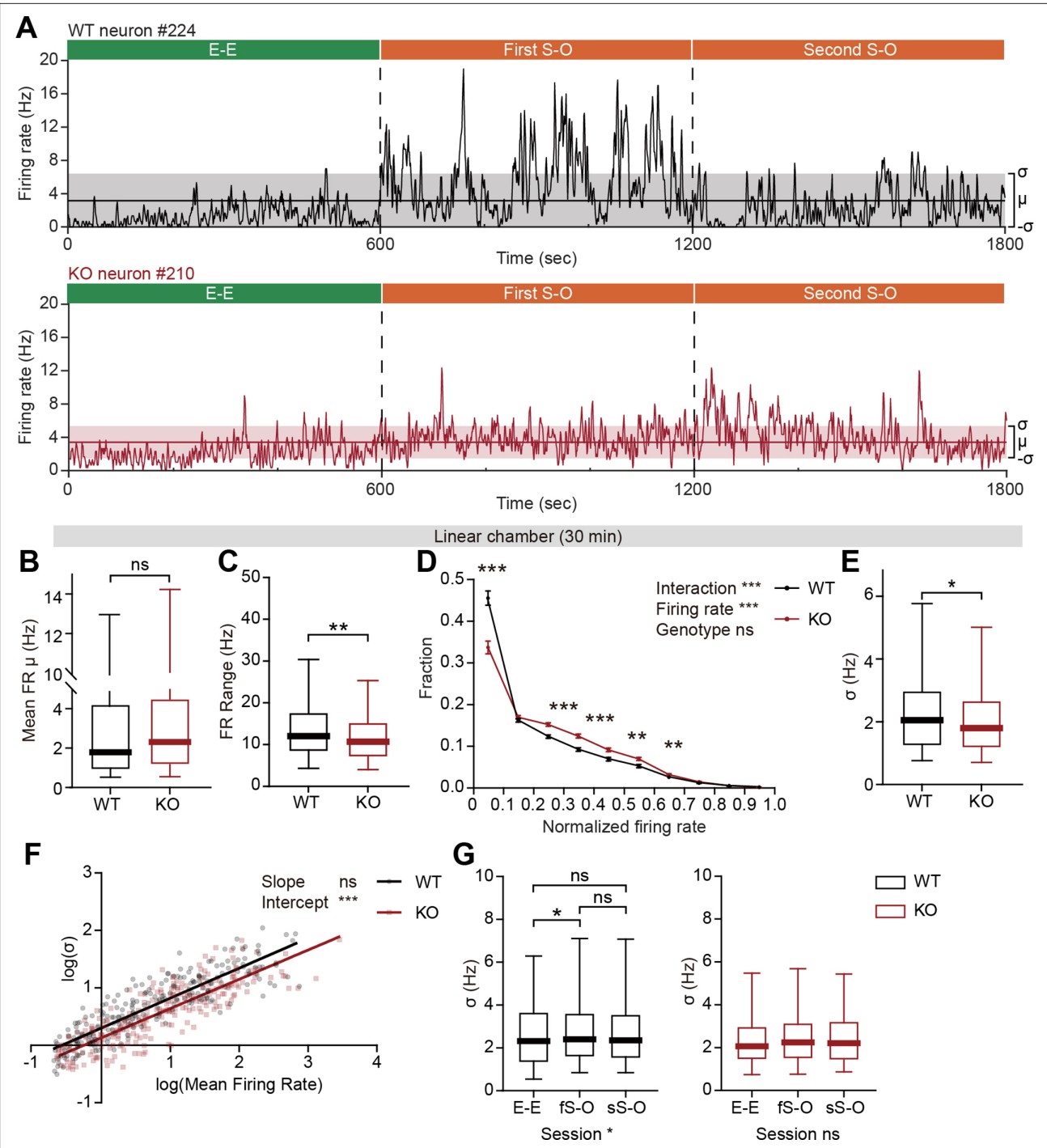

**Figure 5.** Limited firing-rate range and variability in IRSp53-KO pExc mPFC neurons during linear chamber exploration. (**A**) Instantaneous firing-rate traces of representative WT (top) and IRSp53-KO (bottom) pExc neurons (3 s window advanced in 1 s steps) during a sample linear chamber experiment (30 min). Solid horizontal lines indicate the overall mean firing rates (µ). Shaded regions indicate one standard deviation ($\sigma$, sigma). (**B**) Mean firing rate of WT and IRSp53-KO pExc neurons during the 30 min linear chamber test. (n=233 [WT-pExc] and 258 [KO-pExc], ns, not significant, Mann-Whitney test). (**C**) Firing-rate ranges (maximum – minimum instantaneous firing rate) of WT and IRSp53-KO pExc neurons during the linear chamber test (n=233 [WT-pExc] and 258 [KO-pExc], **p<0.01, Mann-Whitney test). (**D**) Mean (± SEM) histograms of normalized instantaneous firing rate during the linear chamber test. For each neuron, instantaneous firing rates were normalized by its maximum instantaneous firing rates. (n=233 [WT-pExc] and 258 [KO-pExc], **p<0.01, ***p<0.001, ns, not significant, two-way RM-ANOVA with Bonferroni's multiple comparisons test). (**E**) Sigma values of the instantaneous firing rates of WT and IRSp53-KO pExc neurons during the linear chamber test. (n=233 [WT-pExc] and 258 [KO-pExc], *p<0.05, Mann-Whitney test). (**F**) Log-scale scatter plot of sigma values against mean firing rates of WT and IRSp53-KO pExc neurons during the linear chamber test. Solid lines indicate

*Figure 5 continued on next page*

*Figure 5 continued*

simple linear regression of WT (black) and KO (red) values. (n=233 [WT-pExc] and 258 [KO-pExc], ***p<0.001, ns, not significant, slope comparison test (see Methods)). (**G**) Sigma values for the instantaneous firing rates of WT (left) and IRSp53-KO (right) pExc neurons during the E-E, first S-O (fS-O), and second S-O (sS-O) sessions of the linear chamber test. (n=233 [WT-pExc] and 258 [KO-pExc], *p<0.05, ns, not significant, Friedman test followed by Dunn's multiple comparisons test). See *Supplementary file 2* for statistics. Numerical data used to generate the figure are available in the *Figure 5—source data 1*.

The online version of this article includes the following source data and figure supplement(s) for figure 5:

**Source data 1.** Source file for instantaneous firing rate data in *Figure 5*.

**Figure supplement 1.** Decreased firing-rate range and variability in IRSp53-KO pExc mPFC neurons selectively during linear chamber exploration.

**Figure supplement 1—source data 1.** Source file for instantaneous firing rate data in *Figure 5—figure supplement 1*.

**Figure supplement 2.** Limited firing-rate changes in response to social and object targets in IRSp53-KO pExc mPFC neurons.

**Figure supplement 2—source data 1.** Source file for firing-rate change data in *Figure 5—figure supplement 2*.

The reduced variability was specific to the recordings from social conditions but not the rest period and non-social conditions (*Figure 5—figure supplement 1I–L*). Similar to firing-rate range, WT neurons showed increased variability in instantaneous firing rate during the first S-O session compared to the E-E session, while IRSp53-KO neurons showed similar levels of variability across the three sessions (*Figure 5G*). The increase in the variability of IRSp53-KO neuronal activity during the first S-O session could not be accounted for by the difference in mean firing rate (*Figure 5—figure supplement 1M*). These results collectively indicate that excitatory mPFC neurons in IRSp53-KO mice have reduced firing-rate range and variability.

## Weak responses to social and object targets in IRSp53-KO pExc mPFC neurons

Limited firing-rate range and variability may indicate limited firing-rate changes in response to social and object targets in IRSp53-KO pExc mPFC neurons. We defined the maximum $\Delta$ firing rate of each session as the maximum absolute difference in mean firing rates between the center zone ($FR_c$) and the two in-zones ($FR_{I1}$, $FR_{I2}$; left and right for E-E session, social and object for S-O sessions) (*Figure 5—figure supplement 2A*).

Compared to WT mice, IRSp53-KO mice displayed a decreased maximum $\Delta$ firing rate only in the first S-O session of the linear-chamber test, but not in the E-E or second S-O session (*Figure 5—figure supplement 2B–D*). IRSp53-KO pExc neurons showed a general decrease in the normalized maximum $\Delta$ firing rate across all three sessions (*Figure 5—figure supplement 2E*). It is notable that the response magnitudes of both WT and IRSp53-KO pExc neurons were the highest during the first S-O session, in response to novel social and object targets. This fits well with our behavioral data, in which only WT mice, but not IRSp53-KO mice, show social preference in the first, but not second, S-O session (*Figure 1D–F*, *Figure 1—figure supplement 2A–C*).

## Impaired burst firing in IRSp53-KO pExc mPFC neurons

Given that firing-rate range and variability is reduced in IRSp53-KO pExc neurons, we reasoned that there may be a shift in the distribution of interspike intervals (ISIs) in IRSp53-KO pExc neurons. Contrary to the comparable levels of average ISI histograms between WT and IRSp53-KO pExc neurons at rest, those during the linear chamber test varied across genotype. In particular, there was a pronounced reduction in the proportion of ISIs ≤10ms in IRSp53-KO pExc neurons (*Figure 6A and B*). This result suggests that the ability to exhibit an abrupt increase in firing rate may be impaired in IRSp53-KO pExc neurons.

Because there was a shift in the ISI distribution of IRSp53-KO pExc neurons, we reasoned that burst firing might be reduced in IRSp53-KO mice. We defined burst spikes as those with short ISIs (≤10ms) (*Figure 6B*). We found that burst firing increased significantly during the linear chamber-exploring state compared to the resting state in WT pExc neurons. Such change, however, was not observed in IRSp53-KO pExc neurons (*Figure 6C*). Burst firing did not differ significantly between WT and IRSp53-KO pExc neurons during the rest period, but was significantly lower in IRSp53-KO than WT pExc neurons during the linear chamber exploration (*Figure 6C*). The same conclusion was obtained when we increased the cut-off value for burst spikes up to 30ms (*Figure 6—figure supplement 1A*

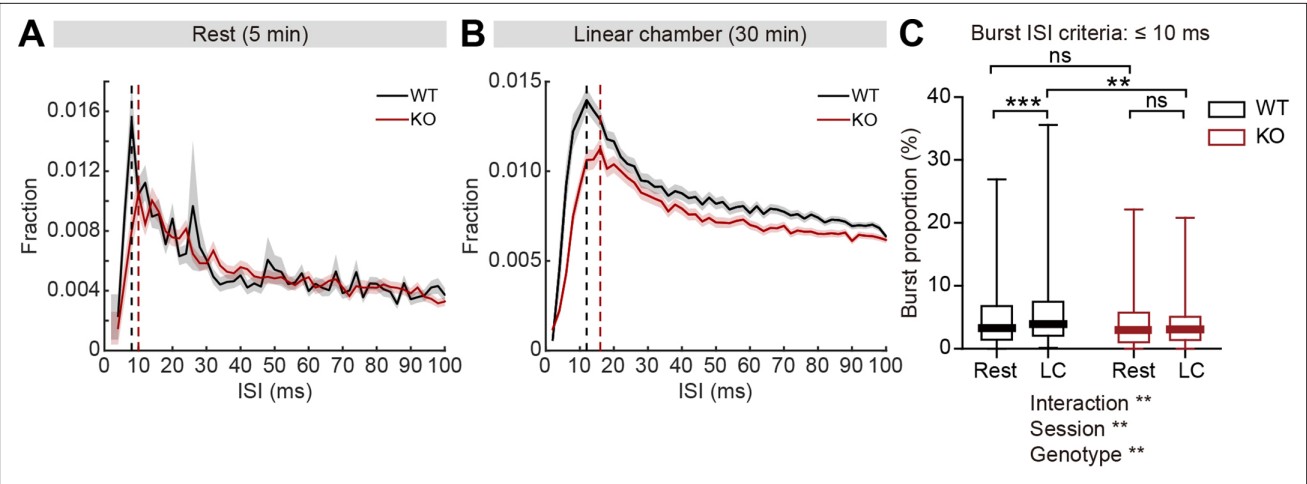

**Figure 6.** Decreased burst firing and spike variability in IRSp53-KO pExc mPFC neurons during linear chamber exploration. (**A and B**) Mean (± SEM) histogram of interspike intervals (ISI) during the 5-min rest (**A**) and the 30 min linear chamber (**B**) periods. Dashed lines denote the peaks of mean ISI distributions for WT (black) and IRSp53-KO (red) neurons (8ms [WT] and 10ms [KO] for rest, 12ms [WT] and 16ms [KO] for linear chamber periods). (n=233 [WT-pExc] and 258 [KO-pExc]). (**C**) Burst proportion (proportion of burst spikes out of total spikes) of WT and IRSp53-KO pExc neurons during the rest and linear chamber (LC) periods for burst ISI threshold of 10ms. (n=233 [WT-pExc] and 258 [KO-pExc], **p<0.01, ***p<0.001, ns, not significant, two-way RM-ANOVA with Sidak's multiple comparisons test). See **Supplementary file 2** for statistics. Numerical data used to generate the figure are available in the **Figure 6—source data 1**.

The online version of this article includes the following source data and figure supplement(s) for figure 6:

**Source data 1.** Source file for ISI and burst firing data in **Figure 6**.

**Figure supplement 1.** Decreased burst firing in IRSp53-KO pExc mPFC neurons during linear chamber exploration.

**Figure supplement 1—source data 1.** Source file for burst firing data in **Figure 6—figure supplement 1**.

**Figure supplement 2.** Burst firing is sufficient for discrimination of social and object targets.

**Figure supplement 2—source data 1.** Source file for firing rate data in **Figure 6—figure supplement 2**.

*and B*). Similar to the firing-rate range, decreased burst firing in IRSp53-KO neurons was observed in both social and non-social conditions (*Figure 6—figure supplement 1C–E*). Taken together, these results indicate that KO pExc neurons show diminished burst firing.

## Burst firing is sufficient for discrimination between social and object targets

To assess the role of burst firing in firing rate variability and social information encoding, we eliminated the burst spikes (ISI ≤10ms) from the spike trains of WT and IRSp53-KO mPFC pExc neurons (*Figure 6—figure supplement 2A*). After burst elimination, the variability, firing-rate range, and response magnitude to social and object targets during the first S-O session remained to be reduced in IRSp53-KO neurons (*Figure 6—figure supplement 2B–E*). The abilities to discriminate between left and right sidedness and social and object targets were also maintained in both WT and IRSp53-KO neurons, although the decoding performance of IRSp53-KO neurons still remained to be lower than that of WT neurons for target discrimination (*Figure 6—figure supplement 2F and G*).

Although these results seem to suggest the unimportance of burst on social information encoding, it is well known that burst spikes increase the probability of post-neuronal spike more effectively than tonic spikes (*Krahe and Gabbiani, 2004*; *Lisman, 1997*), and that simple elimination of burst spikes, post-recording, do not eliminate the real-time effects of burst on neighboring neurons. We therefore alternatively eliminated the tonic spikes (ISI >10ms) to assess whether burst alone is sufficient to support the observed genotypic differences in firing rates (*Figure 6—figure supplement 2H*). The spike variability, firing-rate range, and response magnitude to social and object targets were still reduced in the KO neurons after tonic spike elimination (*Figure 6—figure supplement 2I–L*). In addition, the decoding performance between social and object targets was maintained in both WT and IRSp53-KO neurons, and still lower in the KO neurons (*Figure 6—figure supplement 2N*).

Interestingly, the left versus right side discriminability during the E-E session was maintained by the burst spikes in WT neurons but not in KO neurons, suggesting that burst spikes may also play a role in side discriminability (*Figure 6—figure supplement 2M*). While the reduced burst in IRSp53-KO neurons may affect left versus right side discriminability, tonic spikes may be sufficient for overcoming this weakness. On the other hand, tonic spikes may not be sufficient to overcome the weakened social versus object discriminability in KO neurons.

## Memantine treatment increases burst firing in IRSp53-KO mPFC neurons

A previous study has shown that treatment of memantine, an NMDA receptor antagonist, rescues social deficits in IRSp53-KO mice (*Chung et al., 2015*). We thus tested if memantine could affect burst firing in the IRSp53-KO mPFC using electrophysiological experiments on brain slice preparations. We did not attempt chronic in vivo recordings in awake animals because of the acute nature of memantine treatment involving a short half-life. We targeted layer 5 neurons because mPFC deep layers have been implicated in cognitive and social functions in mouse models of psychiatric disorders (*Brumback et al., 2018*; *Kim et al., 2020a*; *Murugan et al., 2017*; *Paulsen et al., 2022*; *Peixoto et al., 2016*; *Phillips et al., 2019*; *Rapanelli et al., 2021*; *Willsey et al., 2013*; *Yamamuro et al., 2020*; *Yan and Rein, 2022*).

When neuronal firing was induced by current injection in layer 5 pyramidal neurons in the prelimbic mPFC, WT and IRSp53-KO neurons did not display a genotype difference in baseline firing (*Figure 7—figure supplement 1A*). In addition, there were no genotype differences in action potential (AP)-related parameters, including input resistance, AP threshold, AP amplitude, full width at half maximum (FWHM), and afterhyperpolarization (AHP) amplitude (*Figure 7—figure supplement 1B–F*). Interspike interval (ISI) analysis indicated a decreasing tendency of burst firing (ISI <10ms) in IRSp53-KO neurons, as indicated by burst proportion (proportion of spikes in bursts) (*Figure 7—figure supplement 1G and H*), similar to the results from in vivo recording (*Figure 6*).

Upon memantine treatment, WT or IRSp53-KO neurons displayed largely unaltered current-firing curve and AP-related parameters, compared with untreated neurons (*Figure 7B and C*, *Figure 7—figure supplement 1B–F*). However, memantine induced increases in FWHM in both WT and IRSp53-KO neurons and in AP amplitude in IRSp53-KO but not WT neurons (*Figure 7—figure supplement 1D and E*), which may contribute to the increasing tendency of current-firing curve in memantine-treated WT and IRSp53-KO neurons (*Figure 7B and C*).

Importantly, memantine treatment induced opposite changes in burst proportion in WT and IRSp53-KO neurons; memantine decreased burst proportion in WT neurons, whereas it increased burst proportion in IRSp53-KO neurons (*Figure 7D–G*).

These results collectively suggest that memantine treatment does not affect total firing in WT or IRSp53-KO neurons whereas it induces opposite changes in burst firing. Memantine treatment seems to shift the balance more towards burst firing in IRSp53-KO neurons, increasing the reduced burst firing, which may be associated with the normalization of social deficit in IRSp53-KO mice.

## Discussion

The present study investigated abnormalities in social representation and neuronal firing patterns in the mPFC of IRSp53-KO mice. A key finding of the study is that decreased firing rate variability, limited dynamic range, and decreased burst firing in IRSp53-KO putative excitatory mPFC neurons are accompanied by significant decrease in social-responsive neurons and reduced decoding performance discriminating social and non-social cues. Memantine, known to rescue social deficits in IRSp53-KO mice, increases burst firing in IRSp53-KO neurons, whereas it decreases burst firing in WT neurons, without affecting total firing in both WT and IRSp53-KO neurons.

The most salient feature observed herein for IRSp53-KO mPFC neurons is a low proportion of social neurons, but not the other target neurons. This finding suggests that reduced ratios of social/non-social encoding neurons may contribute to social deficits. Similar disruptions in cortical representation of social contexts have been reported in previous studies on mouse models of ASD (*Lee et al., 2021a*; *Lee et al., 2021b*; *Levy et al., 2019*). In addition, weakened response during social exploration is

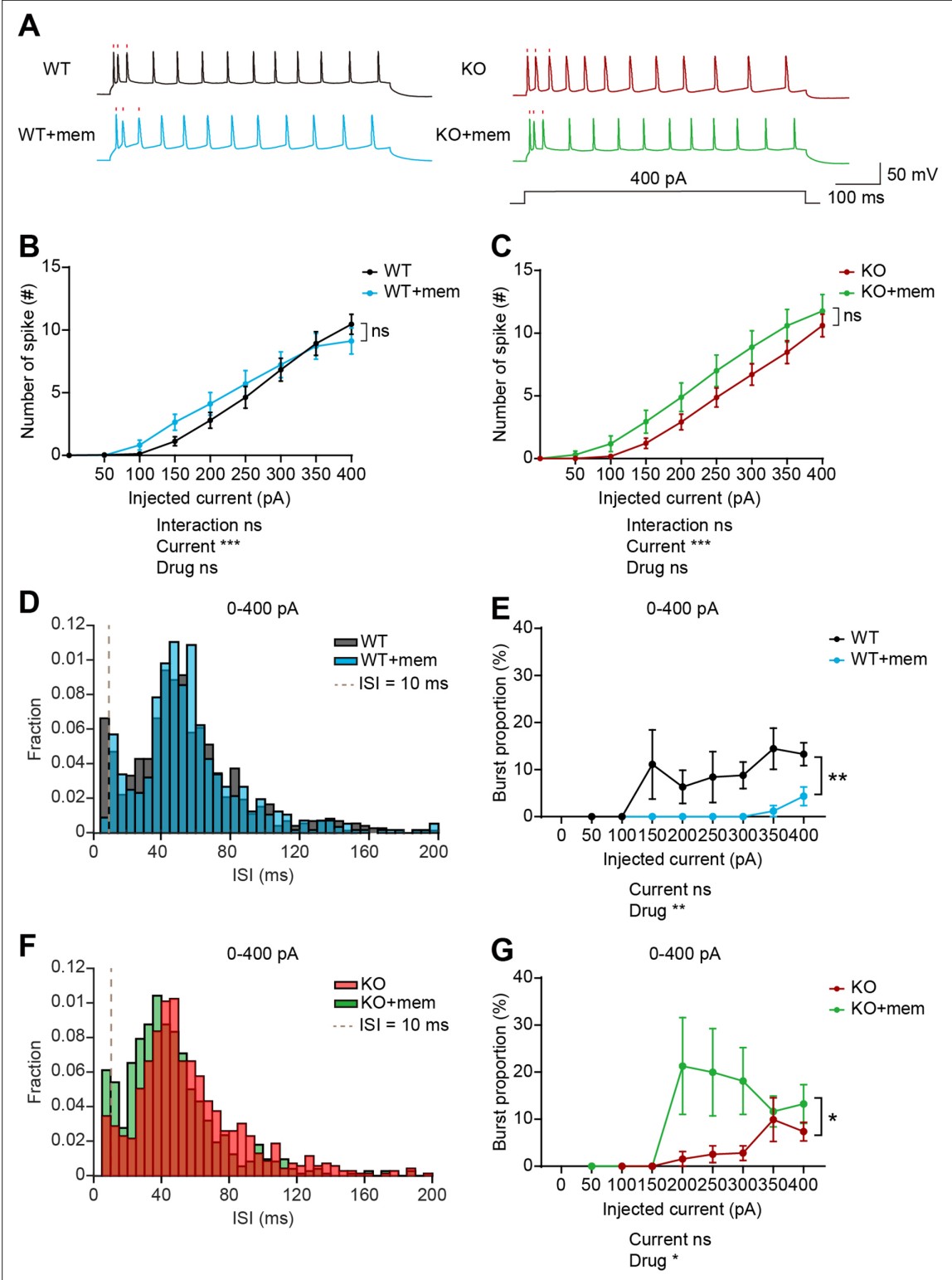

**Figure 7.** Memantine treatment increases and decreases burst firing in WT and IRSp53-KO prefrontal neurons, respectively. (**A**) Examples of firing traces induced by current injections in memantine-untreated and memantine-treated (+mem) WT and IRSp53-KO layer 5 pyramidal neurons in mouse brain slices containing the prelimbic region of the mPFC. Timings of the first three action potentials are indicated by raster plots (red). (**B and C**) Differences in intrinsic excitability between memantine-untreated and memantine-treated layer 5 pyramidal neurons in the prelimbic region of the mPFC in WT (**B**) and IRSp53-KO mice (**C**) (3–6 months), indicated by current-firing curves. (n=24 neurons from 3 mice [WT], 18, 3 [WT +mem], 23, 3 [KO], 17, 3 [KO +mem], ***p<0.001, ns, not significant, two-way RM-ANOVA). (**D and E**) Interspike interval (ISI) histogram (**D**) and burst firing patterns (**E**, current-burst firing

*Figure 7 continued on next page*

*Figure 7 continued*

curve) in layer 5 pyramidal neurons in the mPFC prelimbic region in WT mice (3–6 months) in the presence and absence of memantine treatment. ISIs below the dashed lines (10ms) are classified as burst firing. (n=24, 3 [WT], 18, 3 [WT +mem], **p<0.01, mixed-effects ANOVA). (**F and G**) ISI histograms (**F**) and burst firing patterns (**G**, current-burst firing curve) in layer 5 pyramidal neurons in the mPFC prelimbic region in IRSp53-KO mice (3–6 months) in the presence and absence of memantine treatment. ISIs below the dashed lines (10ms) are classified as burst firing. (n=23, 3 [KO], 17, 3 [KO +mem], *p<0.05, mixed-effects ANOVA). See *Supplementary file 2* for statistics. Numerical data used to generate the figure are available in the *Figure 7— source data 1*.

The online version of this article includes the following source data and figure supplement(s) for figure 7:

**Source data 1.** Source file for slice electrophysiology data in *Figure 7*.

**Figure supplement 1.** Intrinsic excitability-related parameters in memantine-treated/untreated WT and IRSp53-KO prefrontal neurons.

**Figure supplement 1—source data 1.** Source file for slice electrophysiology data in *Figure 7—figure supplement 1*.

observed in mouse models with social deficits (*Brumback et al., 2018*; *Tan et al., 2021*; *Xu et al., 2022*).

The decreased firing-rate range and burst firing in IRSp53-KO mPFC putative excitatory neurons were observed in both social (S-O session) and non-social (E-E session) conditions, although they were not observed in resting conditions. It is conceivable that, as the subject mice are repeatedly exposed to the linear chamber test, the subject mice in the E-E session may remember the interaction with another social target in the past S-O sessions and may expect a social target in advance. Therefore, E-E sessions may have some aspects of social context, which might have been manifested as decreases in firing rate range and burst firing in IRSp53-KO neurons in the E-E session.

IRSp53-KO mPFC putative excitatory neurons display increased firing rates under resting conditions. It is unclear how this increase is caused. A change in the intrinsic excitability is unlikely because baseline firing was not changed in IRSp53-KO mPFC layer 5 neurons (*Figure 7—figure supplement 1*). Although further details remain to be determined, the increased baseline firing in resting conditions might make the mutant neurons 'noisier' or have limited dynamic firing range, disrupting the reliable filtration and transduction of important signals, such as, social and sensory cues, as reported previously for *Cntnap2*-mutant mice (*Levy et al., 2019*) and *Ptchd1*-mutant mice (*Nakajima et al., 2019*), respectively.

Notably, a previous study reports that IRSp53-KO mPFC neurons display decreased firing rates, compared with WT neurons, under urethane anesthesia conditions (*Chung et al., 2015*). Brain circuits are known to be more active during the awake state, compared to an anesthetized state (*Albrecht and Davidowa, 1989*; *Shumkova et al., 2021*). In addition, urethane has diverse effects on neurotransmitter receptors (GABA, glycine, acetylcholine, NMDA, and AMPA receptors) and neuronal excitability (*Accorsi-Mendonça et al., 2007*; *Hara and Harris, 2002*; *Sceniak and Maciver, 2006*). Given the broad effects of urethane, it would be difficult to narrow down and specify the key mechanism underlying the difference between awake and anesthetized states in IRSp53-KO mice.

The fact that social deficits are induced by IRSp53 deletion in mice may help us better understand social-related neural mechanisms at multiple levels, including actin dynamics at excitatory synapses, excitatory and inhibitory synaptic balance, NMDAR modulation, and burst firing. IRSp53 is a signaling adaptor and scaffolding protein that regulates excitatory synaptic actin dynamics and dendritic spine morphogenesis (*Kang et al., 2016*). IRSp53/BAIAP2-related mutations in humans have been implicated in ASD, ADHD, and schizophrenia. IRSp53 deletion in mice alters excitatory synapse number and function in the prefrontal cortex, hippocampus, and amygdala (*Bobsin and Kreienkamp, 2016*; *Chung et al., 2015*; *Kim et al., 2009*; *Kim et al., 2020b*; *Sawallisch et al., 2009*). IRSp53 deletion also abnormally strengthens excitatory synaptic actin filaments and NMDAR activity through impaired long-term depression of NMDARs (*Chung et al., 2015*). Intriguingly, inhibiting the NMDAR hyperactivity by memantine treatment rapidly normalizes the social deficits in IRSp53-KO mice (*Chung et al., 2015*). These results suggest that (1) actin modulatory proteins could have strong impacts on NMDAR function, supporting the reported link between dysregulated actin and mental disorders (*Yan et al., 2016*), (2) excessive NMDAR function could lead to social deficits, corroborating the NMDAR dysfunction hypothesis for ASD (*Lee et al., 2015a*), (3) altered balance of excitatory and inhibitory synaptic transmission could lead to social deficits, supporting the excitation/inhibition imbalance hypothesis for ASD (*Lee et al., 2017*; *Nelson and Valakh, 2015*; *Sohal and Rubenstein, 2019*; *Yizhar et al.,*

*2011*), and (4) NMDAR modulators have a therapeutic potential for social dysfunction-related psychiatric disorders.

Our current results extend these findings by proposing that decreased firing variability and burst firing may connect NMDAR dysfunction with prefrontal social encoding and behavioral social deficits in IRSp53-KO mice. In support of this hypothesis, memantine, which rescues social deficits in IRSp53-KO mice, normalizes the decreased burst firing in mPFC neurons. Burst firing has been implicated in stronger presynaptic release, reliable synaptic transmission, synaptic plasticity, information processing, and neuropeptide release (*Cui et al., 2019*; *Krahe and Gabbiani, 2004*; *Lisman, 1997*; *Payeur et al., 2021*). In addition, burst firing can be regulated by NMDARs in various types of neurons (*Clark and Chiodo, 1988*; *Cui et al., 2019*; *Grienberger et al., 2014*; *Jackson et al., 2004*; *Lee et al., 2021a*; *Zweifel et al., 2009*). In addition, Shank2-KO mice with NMDAR hypofunction display social deficits accompanying impaired prefrontal social representation that are responsive to pharmacological NMDAR activation or optogenetic burst activation (*Lee et al., 2021a*; *Lee et al., 2015b*; *Won et al., 2012*).

However, it remains unclear how IRSp53-KO prelimbic neurons show impaired target-induced burst firing (*Figure 6*) despite that their NMDAR function is increased in layer 5 prelimbic neurons (*Noh et al., 2022*), and how acute memantine treatment increases and decreases burst firing in WT and IRSp53-KO neurons, respectively (*Figure 7F and G*). Memantine is known to act on extrasynaptic NMDARs but not on synaptic NMDARs at low concentrations, which was found to be useful for suppressing the neurotoxic effects of extrasynaptic NMDAR activation in Alzheimer's and Huntington's diseases (*Hardingham and Bading, 2010*; *Okamoto et al., 2009*; *Xia et al., 2010*). It is therefore possible that the amount and balance of synaptic and extrasynaptic NMDARs might have been altered in IRSp53-KO mice in a way that memantine acts more strongly on extrasynaptic NMDARs relative to synaptic receptors, leading to strong memantine-induced disinhibition of synaptic NMDARs, whereas, in WT synapses, memantine acts more strongly on synaptic NMDARs relative to extrasynaptic NMDARs. This difference might involve excessive accumulation of NMDARs at IRSp53-KO synaptic sites, caused by abnormal actin stabilization and limited LTD of NMDARs (*Chung et al., 2015*), which might induce extrasynaptic accumulation of NMDARs. This might induce suppressed burst firing in mutant neurons and enhanced burst firing in memantine-treated mutant neurons, given the reported synergistic effects between synaptic NMDARs and burst firing (*Lee et al., 2021a*; *Yang et al., 2018*).

It should be pointed out, however, that because neural activity and behavior occurs in a simultaneous manner, it remains unclear whether the changes in neural activity cause social deficit, or rather, social deficit results in changes in neural activity. For instance, we do not know whether the reduced proportion of social encoding neurons represents the cause of limited social brain functions, or rather, is an outcome of reduced social interaction or even limited sensory input. In addition, although our research focuses on the mPFC, we cannot rule out the possibility that the differences in firing in IRSp53-KO pExc mPFC neurons may have stemmed from deficits other brain regions.

In summary, our results indicate that IRSp53-KO mice display elevated spontaneous firing and reduced firing variability and burst firing in mPFC neurons, which may suppress cortical social representation and induce behavioral social deficits. In addition, memantine that can rescue social deficits in IRSp53-KO mice improves reduced burst firing in mPFC neurons, suggesting that burst firing may link NMDAR dysfunction with social deficits.

# Materials and methods

**Key resources table**

| Reagent type (species) or resource | Designation | Source or reference | Identifiers | Additional information |
|---|---|---|---|---|
| Gene (*Mus musculus*) | IRSp53 (*Baiap2*) | *Kim et al., 2009* | N/A | |
| Software, algorithm | DeepLabCut | *Lauer et al., 2022*; *Mathis et al., 2018* | Ver 2.0, 2.2 | Used for tracking of mice body parts. |
| Software, algorithm | Ethovision XT 13 | Noldus https://www.noldus.com/ | Ver 13 | Used for measuring the locomotion of mice. |
| Software, algorithm | GraphPad Prism 9.0 | GraphPad https://www.graphpad.com/ | Ver 9.0 | Used for all statistics used in the current study. |

*Continued on next page*

*Continued*

| Reagent type (species) or resource | Designation | Source or reference | Identifiers | Additional information |
|---|---|---|---|---|
| Software, algorithm | MATLAB | MathWorks https://www.mathworks.com/ | Ver 2020a | Used for spike analysis. |
| Software, algorithm | MClust 4.0 | David Redish http://redishlab.neuroscience.umn.edu/MClust/MClust.html | Ver 4.0 | Used for manual single-unit isolation. |
| Software, algorithm | Clampex | Molecular devices | Ver 10.7 | Used for recording electrophysiological signals from cells |
| Software, algorithm | Clampfit | Molecular devices | Ver 10.7 | Used for analysing electrophysiological data |
| Software, algorithm | pCLAMP | Molecular devices | Ver 10 | Used for recording electrophysiological signals from cells |
| Software, algorithm | IntrinsicVIEW | https://github.com/parkgilbong/IntrinsicVIEW; *Kim and Kim, 2020* | N/A | Used for intrinsic property analysis |
| Other | Digital lynx SX | Neuralynx https://neuralynx.com/ | | Used for in vivo single-unit recording |
| Other | Multiclamp 700B | Molecular devices | | Used for recording electrophysiological signals from cells |
| Other | Digidata 1550 | Molecular devices | | Used for recording electrophysiological signals from cells |
| Chemical compound, drug | Memantine hydrochloride ≥98% (GC) | Sigma | M9292 | Used for slice electrophysiology experiment |

## Animals

Adult (3–6 months old) C57B/6 J male WT (n=6) and IRSp53-KO (n=8) mice were used for single-unit recording and slice electrophysiology. Mice were fed ad libitum and maintained under 12 hr light/dark cycle (light period 1 am–1 pm). All experiments were conducted during the dark phase (1 pm–1 am) of the light/dark cycle. The mouse facility and experimental setting were always maintained at 21 °C and 50–60% humidity. Mice were maintained according to the Animal Research Requirements of KAIST. All experiments were conducted with approval from the Committee on Animal Research at Korea Institute of Science and Technology (approval number KA2020-94).

## Linear-chamber social-interaction test

Before linear-chamber social-interaction test (*Lee et al., 2016*), the subject mouse was placed in a white 7.5 cm radius x 15 cm opaque acryl container for neural recording at rest for 5 min. The mouse was then allowed to explore the 45 cm x 10 cm x 21 cm linear chamber with empty-empty chambers (E-E session), social and object chambers (first S-O session), and then the same object-social chambers with the side exchanged (second S-O session), for 10 minutes each. The social and object targets used for each experiment were always novel. Novel male 129/Sv mice of similar age were used as the social target. For the first S-O session, the placement of social and object targets into left or right chambers were randomly chosen. All recordings were conducted at 30 lux. A total of 12 recording experiments were conducted for each mouse, with 3 days of isolation interval.

## Mice movement tracking

Mice movements were monitored by a digital camera mounted on the ceiling, directly above the linear chamber assay. The position of the subject mice (nose, right and left ears, body center, and tail-base), social target (nose and body center), and four corner points of the linear chamber were trained using a pose estimation software DeepLabCut (version 2.0 *Mathis et al., 2018* and 2.2 *Lauer et al., 2022*).

## Behavioral analysis

Sniffing time was defined as the time when the nose point is within 3 cm from the face of each target chamber. In-zone time was defined as the time when the body center (midpoint between nose point and tail base) is within 9 cm from the face of each target chamber. Social and object target interactions were further classified into proximal and distal interactions. Social proximal interaction was when the

distance between the noses of the subject mouse and the social target was within 0 and 2.5 cm, while social distal interaction was when the nose-to-nose distance was within 2.5 and 10 cm. For object interaction, the distance between subject mouse's nose and the center of object chamber face was assessed.

The distance moved was based on the body center of the subject mice and automatically measured via video tracking software, Ethovision XT 13 (Noldus Information Technology).

### Single-unit recording

Eight tetrodes were implanted in the mPFC (four tetrodes per hemisphere; 1.7–2.1 mm anterior and 0.1–0.5 mm lateral from bregma, and 1.5–2.3 mm ventral from brain surface). 36-channel electrode interface board (EIB-36; Neuralynx, Bozeman, MT, USA) and hyperdrive (modified version of Flex drive from Open Ephys) were used. Mice were subjected to 3 days of handling (10 min each day) after 1 week of recovery from surgery. At first exposure to the linear chamber test, mice were habituated to the environment and the tether but without recording. After habituation, mice were subjected to 12 linear-chamber experiments with recording. 10,000 x amplified single-unit recording signals with 32 kHz sampling frequency were filtered using a bandpass filter of 600–6000 Hz. Signals were recorded via Digitalynx (hardware; Neuralynx, Bozeman, MT, USA) and Cheetah data-acquisition system (software version 5.0; Neuralynx, Bozeman, MT, USA) and stored in a personal computer. In order to record different units at each recording experiment, the positions of tetrodes were lowered by 62.5 µm after the recording.

### Histology

After the 12th recording, mice were deeply anesthetized and the locations of the tetrodes were marked by electrolytic lesion (100 µA unipolar current for 7 s for each electrode) and brains were extracted and perfused in 4% Paraformaldehyde (PFA) solution for at least 72 hr. The fixed brains were sliced coronally (50 µm) using a vibratome (VT1000; Leica, Buffalo Grove, IL, USA), stained with DAPI, and the positions of lesions were assessed by post hoc histological evaluation using a confocal microscope (LSM780; Carl Zeiss, Oberkochen, Germany).

### Spike analysis

Single-unit spike clusters were isolated manually by spike waveform features, such as, energy, peak, valley, and principal components, using MClust (version 4.4, available online at http://redishlab.neuro-science.umn.edu/mclust/MClust.html; credits to A. David Redish). Only units with isolation distance above 25 and L-ratio below 0.1 were used for analysis.

Only valid sniffing trials, valid in-zone (and center zone) trials, and valid proximal and distal trials were used for spike analysis. Valid sniffing trials were defined as those with a duration of ≥1 s and inter-trial interval of ≥2 s. For sufficient acquisition of center zone trials, valid in-zone (and center zone) trials were defined as those with a duration of ≥0.5 s and inter-trial interval of ≥0.5 s. Valid proximal and distal trials were defined as those with a duration of ≥1 s. Neurons with missing valid sniffing and in-zone trials for any of the six targets (left-right for the E-E session, social-object for the first and second S-O sessions) were excluded from spike analysis. The number of neurons recorded after valid trial exclusion was WT n=391 total neurons, 366 pExc neurons, 17 pInh neurons from 6 mice and IRSp53-KO n=394 total neurons, 359 pExc neurons, 24 pInh neurons from 8 mice (see *Supplementary file 1* for details). The pExc and pInh neurons were classified based on half-valley width (pExc: HVW >200ms, pInh: HVW <200ms) and peak to valley ratio (pExc: PVR >1.4, pInh: PVR <1.4).

Except for the comparison of mean firing rate at rest (total number of spikes within 5 min resting duration), only pExc neurons with the average firing rate of ≥0.5 Hz during the 30 min linear chamber assay were used for further analysis. After ≥0.5 Hz filtration, the total number of pExc neurons were WT n=233 neurons from 6 mice and IRSp53-KO n=258 neurons from 8 mice (see *Supplementary file 1* for details).

### Target (empty, social, object) neuronal proportion analysis

For each neuron, the valid sniffing bouts towards each of empty (E), social (S), and object (O) targets were divided into 0.5 s time-bin and its instantaneous firing rates were calculated. The instantaneous firing rates at the center zone across the E-E, first S-O and second S-O sessions were also calculated.

To calculate the area under the receiver operating characteristic curves (auROCs), we generated histograms comparing the distribution of instantaneous firing rates across target sniffing trials and the distribution of instantaneous firing rates at the center zone (*Britten et al., 1992*; *Li et al., 2017*). The target sniffing and center zone firing rate distributions were compared by moving a threshold from minimum to the maximum firing rate (x-axis of the histogram). The range of threshold was divided into 100 bins. ROC curves were generated by plotting the probability that the target sniffing firing rate was greater than the threshold against the probability that the center zone firing rate was greater than the threshold. The calculated auROCs (0–1.0) indicate the response of each neuron to targets relative to the center zone. Neurons with less than 40 instantaneous firing rates (i.e. shorter than 20 s) for any targets or center zone were removed from analysis.

We also randomly shuffled these target sniffing and center zone instantaneous firing rates and then calculated the shuffled auROC for 1000 times. A neuron was classified as target responsive neuron if its actual auROC value was greater than the 99.5% quantile or lower than the 0.5% quantile of the distribution of its shuffled auROCs.

For generating spike density functions (SDFs), firing rates of each neuron were calculated in 250 ms bins (from –1.5 sec to 3 s after the onset of target sniffing) and averaged across the sniffing trials. For the averaged SDF of target neurons, the averaged firing rates of individual neurons were normalized by their maximum firing rate, and then averaged across all target neurons.

### Target neuron consistency analysis

To analyze the consistency of neuronal response to the targets across sessions, the z-scores for social and object targets and left and right sidedness in the first S-O session were plotted against those in the second S-O session and their correlation were tested. The z-scores for each neuron were defined as $(FR_T - \mu_C)/\sigma_C$, in which $FR_T$ is the mean firing rate during sniffing of each target (E, S, or O), while $\mu_C$ and $\sigma_C$ are the mean and one standard deviation (SD) of the instantaneous firing rates at the center zone, respectively.

For examining the effect of proximity on magnitude of firing rate response to the targets, the absolute z-scores during proximal and distal interactions were calculated for both S and O targets, normalized to the firing rate at the center zone.

Trial-to-trial consistency (%) of neuronal response to targets was calculated for social and object neurons. For each target neuron, z-scores were calculated for each sniffing trial (from onset of interaction to the end). Trial consistency of each increasing and decreasing target neuron was evaluated through calculating the percentage of trials with a positive or negative z-score, respectively.

### Neural decoding of target via the support vector machine (SVM)

Single and population neuronal decoding of targets were assessed by SVM classifier. For individual neuron analysis, 10 left and 10 right sniffing trials (for E-E session) or 10 social and 10 object sniffing trials (for S-O session) were randomly selected. We adopted leave-one-out cross validation, in which a single trial was removed from each of the two groups (left and right groups or social and object groups) and tested for prediction. Based on the firing rates of the remaining 18 sniffing trials (training trials), we predicted which side (left or right) or target (social or object) the 2 removed trials (test trials) belong to. Each of the 10 trials was removed one at a time for prediction. For each neuron, this process was repeated 100 times, and the average prediction accuracy (%) was calculated. Correct neurons were defined as neurons with a decoding accuracy >55% (above chance level).

For population decoding, given number of neurons were randomly selected according to the ensemble size and their trials were combined for decoding analysis. This process of random selection of neurons was repeated 100 times to derive 100 decoding accuracy values per ensemble size. Since at least 10 sniffing trials were required for each group, neurons with less than 10 sniffing trials for any of left and right sidedness or social and object targets were removed from analysis. We have modified an open source code (*Bae et al., 2021*), which uses MATLAB 'fitcsvm' function for SVM decoding.

### Instantaneous firing rate and firing rate variability analysis

For instantaneous firing rate analysis, 30 min linear chamber sessions were divided into 1800 time-bins (3 s time-bin of 1 s steps). Firing rate range is the difference between maximum and minimum instantaneous firing rate. The normalized instantaneous firing rate of each neuron was calculated by

dividing each instantaneous firing rate by the maximum instantaneous firing rate. Sigma (Hz) for each neuron was defined as 1 standard deviation (1SD; includes 68% of data) value of the 1800 instantaneous firing rates.

## Maximum Δ firing rate analysis

We divided the linear chamber into five equal-area sections (each 9 cm in length), and the firing rates in in-zone areas were compared to that of the center zone. Maximum Δ firing rate of a neuron is the maximum value between absolute firing rate differences between the firing rates at two in-zones and the firing rate at center zone (($|FR_{I1} - FR_C|$ and $|FR_{I2} - FR_C|$) where $FR_c$ is the firing rate at the center zone and $FR_{I1}$ and $FR_{I2}$ are the firing rates at two in-zones). The two in-zones are left and right in the E-E session, and social and object in the first and second S-O sessions.

The normalized maximum Δ firing rate is the maximum value between absolute normalized firing rate differences between the firing rates at two in-zones and the firing rate at the center zone ($|(FR_{I1} - FR_C)/(FR_{I1} + FR_C)|$ and $|(FR_{I2} - FR_C)/(FR_{I2} + FR_C)|$).

## Burst analysis

Interspike interval (ISI) is the time between two consecutive spikes (in ms). For the average ISI histogram, ISI ≤200ms were extracted for each neuron, and the ISI histogram values of individual neurons were averaged.

Burst proportion (%) was defined as the number of burst spikes out of total spikes of a neuron, in which burst spikes are defined as consecutive spikes with ISI ≤10ms. To demonstrate the cut-off effects of ISI burst definition, burst analysis was performed using a range of burst ISI threshold values (5–30ms).

For burst elimination, two spikes that make up ISIs ≤10ms were all eliminated. For tonic elimination, all spikes other than the two spikes that make up ISIs ≤10ms were all eliminated.

## Slice electrophysiology

Mice were anesthetized with isoflurane and decapitated. After brain dissection, coronal slices of medial prefrontal cortex (300 μm) were cut in ice-cold oxygenated sucrose-based cutting solution containing (in mM): 75 Sucrose, 76 NaCl, 2.5 KCl, 25 NaHCO₃, 25 Glucose, 1.25 NaH₂PO₄, 7 MgSO₄, 0.5 CaCl₂ with pH 7.3, and 310 mOsm by using vibratome 7000smz-2 (Campden instruments, England), and then recovered in the same solution for 30 min at 33–34°C. Slices were then transferred to an incubation chamber filled with oxygenated artificial cerebral spinal fluid (ACSF) containing (in mM): 124 NaCl, 2.5 KCl, 1.3 MgCl₂, 2.5 CaCl₂, 1.0 NaH₂PO₄, 26.2 NaHCO₃, 20 Glucose with pH 7.4 and 310–313 mOsm at room temperature and slices were kept in less than 7 hr before recordings.

For patch-clamp recording, slices were transferred to a recording chamber perfused with oxygenated ACSF at 30–32 °C controlled by a peristaltic pump. Neurons in prelimbic region in mPFC were visualized with ×60 magnification objective (Olympus) simultaneously on the stage of upright microscope (BX51W1, Olympus). Patch microelectrodes were pulled from borosilicate glass (O.D.:1.5 mm, I.D.: 1.10 mm, WPI) on a Flaming-Brown micropipette puller model P-1000 (Sutter Instruments, USA). Patch microelectrodes had a resistance of 4–8 MΩ. Signals were recorded using a patch-clamp amplifier (Multiclamp 700B, Axon Instruments, USA) and digitized with Digidata 1550 A (Axon Instruments, USA) using Clampex software. Signals were amplified, sampled at 10 kHz, and filtered to 2 or 5 kHz. Pyramidal neurons in layer 5 were identified by large apical dendrites.

During the current- and voltage-clamp recording, the membrane potential was maintained at −70 mV. Neurons in which the holding current exceeded 500 pA were excluded from the analysis.

In current-clamp recordings, membrane potential was held at −70 mV with intracellular solution (in mM): 135 K-gluconate, 7 NaCl, 10 HEPES, 0.5 EGTA, 2 Mg-ATP, 0.3 Na₂-GTP, 10 Na-phosphocreatine with pH 7.3 and 295–297 mOsm. Current-clamp experiments were recorded 5 min after obtaining whole-cell configuration.

For intrinsic excitability recording, Hyperpolarization currents and action potentials (APs) were generated by injecting 500ms current steps from −300–400 pA increasing by 50 pA. To exclude synaptic inputs, synaptic blockers were bath applied during slice recording: GABAzine (SR 95531) 10 μM for GABAa receptor, APV 50 μM for NMDA receptor, NBQX 10 μM for AMPA receptor.

To validate the effect of memantine on NMDA receptors in the pyramidal neurons, memantine 1 μM was bath applied with brain slices for at least 30 min. During bath application of memantine for over 10 min, if the holding current of a neuron exceeded 500 pA or the series resistance (Rs) was altered by 15%, that neuron was discarded from the analysis.

## Analysis

Current-clamp recordings were analyzed with IntrinsicView (LabVIEW program made available at https://github.com/parkgilbong/IntrinsicVIEW; *Kim, 2022 Kim and Kim, 2020*). To evaluate intrinsic excitability, 500 ms-long depolarizing currents were injected from –300–400 pA with increments of 50 pA, and the mean firing rate was calculated based on the number of evoked APs in response to a depolarizing current injection. The input resistance ($R_{in}$) was determined by measuring the difference between the baseline and the steady-state (post-sag) $V_m$ deflection generated by –100 pA square current. The voltage threshold of AP was defined by measuring the membrane potential at which its 1st derivatives exceeded 5 mV/ms. The differences between the AP threshold and the positive and negative peaks of the trace were defined as the AP amplitude and the AHP amplitude, respectively. The FWHM was defined as the time difference between two points that is equal to the half of the amplitude.

For burst analysis, every two or more spikes that make up ISIs ≤10ms were classified as burst spikes. Burst proportion (%) was calculated for each neuron for every depolarizing currents and statistically compared between genotypes.

## Statistical analysis

Statistical significance was determined via repeated measures of Two-Way ANOVA with Sidak's (or Bonferroni's or Tukey's) multiple comparisons test, Friedman test with Dunn's multiple comparisons test, Mann-Whitney test, unpaired *t*-test, one sample *t*-test, simple linear regression with slope comparisons test, Chi-square test, and mixed-effects ANOVA (all via Prism 9.0; GraphPad, San Diego, CA, USA). Kolmogorov-Smirnov normality test was used to determine whether to use a parametric or nonparametric test. Graphs were generated by MATLAB 2020a (MathWorks; Natick, MA, USA) and Prism 9.0. All box and whisker plots show median, interquartile range, and 2.5 and 97.5 percentile. All violin plots show median and interquartile range. See *Supplementary file 2* for details on statistics and **source files** for raw data.

## Acknowledgements

We thank Changho Jo and Seohui Bae for their help with DeepLabCut. This study was supported by the National Research Foundation of Korea (NRF) grants funded by the Korean Government (MSIT) (NRF-2022M3E5E8018388 to EL) and the Ministry of Health & Welfare, Republic of Korea (KHDI *HI21C1659* to EL), IBS-R002-D2 (to MWJ) and IBS-R002-D1 (to EK).

## Additional information

### Competing interests

Eunjoon Kim: Reviewing editor, *eLife*. The other authors declare that no competing interests exist.

### Funding

| Funder | Grant reference number | Author |
| --- | --- | --- |
| National Research Foundation of Korea | NRF-2022M3E5E8018388 | Eunee Lee |
| Ministry of Health and Welfare | KHDI HI21C1659 | Eunee Lee |
| National Research Foundation of Korea | NRF-2019R1A2C4069863 | Se-Bum Paik |
| Institute for Basic Science | IBS-R002-D2 | Min Whan Jung |

| Funder | Grant reference number | Author |
| --- | --- | --- |
| Institute for Basic Science | IBS-R002-D1 | Eunjoon Kim |

The funders had no role in study design, data collection and interpretation, or the decision to submit the work for publication.

## Author contributions

Woohyun Kim, Conceptualization, Data curation, Formal analysis, Investigation, Visualization, Writing - original draft, Writing - review and editing; Jae Jin Shin, Formal analysis, Investigation; Yu Jin Jeong, Jung Won Bae, Formal analysis; Kyungdeok Kim, Young Woo Noh, Investigation; Seung-joon Lee, Visualization, Methodology; Woochul Choi, Resources; Se-Bum Paik, Resources, Funding acquisition; Min Whan Jung, Formal analysis, Funding acquisition, Writing - review and editing; Eunee Lee, Conceptualization, Formal analysis, Funding acquisition, Methodology, Writing - review and editing; Eunjoon Kim, Supervision, Funding acquisition, Project administration, Writing - review and editing

## Author ORCIDs

Woohyun Kim http://orcid.org/0000-0002-9820-4401
Kyungdeok Kim http://orcid.org/0000-0002-0003-6957
Young Woo Noh http://orcid.org/0000-0001-8280-2717
Se-Bum Paik http://orcid.org/0000-0002-4078-305X
Min Whan Jung http://orcid.org/0000-0002-4145-600X
Eunee Lee http://orcid.org/0000-0001-9726-4819
Eunjoon Kim http://orcid.org/0000-0001-5518-6584

## Ethics

Mice were maintained according to the Animal Research Requirements of Korea Advanced Institute of Science and Technology (KAIST). All experiments were conducted with approval from the Committee on Animal Research at KAIST (approval number KA2020-94).

## Decision letter and Author response

Decision letter https://doi.org/10.7554/eLife.74998.sa1
Author response https://doi.org/10.7554/eLife.74998.sa2

## Additional files

### Supplementary files

- Supplementary file 1. Recorded number of neurons from each WT and IRSp53-KO mouse.
- Supplementary file 2. Statistics for all figures provided.
- Transparent reporting form

### Data availability

Source Data files have been provided for all figures (except for Figure 1 - figure supplement 1, Figure 3, and Figure 3 - figure supplement 1).

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
