## [Editor Report]

This study by Kim et al. is of interest to neuroscientists studying neocortical neural activity, as related to social behavior and in mouse models of neuropsychiatric disorders. These results provide new data on how the loss of the postsynaptic scaffolding and adaptor protein IRSp53 impacts prefrontal cortex activity and social interaction in mice. The authors propose the interesting idea that suppressed neuronal activity dynamics and burst firing may contribute to the impaired cortical encoding of social information and social behaviors in IRSp53-mutant mice.

---

## [Decision Letter]

**Decision letter after peer review:**

Thank you for submitting your article "Suppressed prefrontal neuronal firing variability and impaired social representation in IRSp53-mutant mice" for consideration by *eLife*. Your article has been reviewed by 3 peer reviewers, and the evaluation has been overseen by a Reviewing Editor and Catherine Dulac as the Senior Editor. The reviewers have opted to remain anonymous.

Essential revisions:

1. More evidence of social impairments in IRSp53 KO mice. Based on the results of Figure 1, the authors stated that KO animals 'display social impairments', but this may not be the only interpretation of their data. In Figure 1E, time spent sniffing the objects during the first S-O session is not significantly different between wild-types and KOs, although variability is higher in KOs; similarly, social sniff times are also variable. Perhaps the KO animals have somewhat stronger sniffs, or are faster at processing the olfactory information, requiring them to spend less time sniffing and remaining in the in-zone. Please address this comment by providing further data demonstrating social impairments in KO mice.

2. Stronger evidence that the observed changes in firing rate really reflect differences in response to social and non-social targets in WT and IRSp53 KO mice. In the identified single-units in the mPFC of either WT or IRSp53 KO, only 10% were responsive to social and/or object targets, while 90% of the recorded units are not responsive to either target (Figure 7E). Consistently, the mean firing rate in E-E, fS-O and sS-O are similar in both WT and KO mice (Figure 3-supplement 1E), indicating that the presence of social targets had little effect in regulating most of the recorded mPFC neurons. Therefore, it is not convincing that the recorded mPFC neurons play an essential role in discriminating social targets. Please examine whether the FR range and burst firing proportion are different between WT and KO in resting and non-social conditions.

3. Suggestions for further data analysis. Perhaps moving the classification in Figure 7 to earlier in the manuscript, and analyzing firing rate statistics separately for 'social' vs 'non-social' neurons. Does unit classification (i.e., of 'social' or not) hold up across sessions or episodes of engagement within a session? If these really are 'social' units, presumably that aspect should be reliable across interactions with different mice.

4. Further analysis of firing rate data. To what extent are changes in the max firing rate during social task / the discrimination index explained by changes in bursting explain all of this? If one takes out the bursts, e.g., eliminate bursts in the spike trains – do changes in the discrimination index, etc. go away? Please also assess the proportion of social and object neurons in shuffled data to rule out the possibility that changes in the proportions of these neurons is due to nonspecific changes in activity. Other suggestions for further analysis include looking at single cell metrics, e.g., area under the receiver operator curve, mutual information, etc.

5. Cleaner analysis relating the unit spike trains to moment-to-moment features of social interaction or other behaviors occurring during the S-O sessions. The authors use machine learning to classify mouse part position, but only seem to mark where the mouse is in the track. The unit activity over time might be much more interesting if correlated with DeepLabCut-based analysis of what the animal was doing (or perhaps what the social interaction partner was doing) during the sessions. Please correlate behavior to activity, both of individual units and of simultaneously-recorded populations.

6. The authors found that the average resting firing rates of excitatory mPFC neurons in awake IRSp53-KO mice are larger than that in WT mice (Figure 1C and 1D). These results are opposite to their previous findings obtained from anaesthetized mice (Chung et al., 2015, Figure 8b and 8c). However, no such difference was observed between awake and anaesthetized WT animals. Please discuss.

7. Additional data on the contribution of NMDA receptors to observed changes in firing, Additional data on the effects of NMDAR antagonists could significantly strengthen the manuscript by potentially providing some mechanistic information and strengthening the correlation between changes in these physiological measures and changes in behavior. Please also provide some discussion about how increased NMDAR function or other neurobiological mechanisms cause decreased bursting.

8. Further discussion on social behavior deficits in IRSp53 KO mice and how components of excitatory synapses contribute to autism. While there may be few autism patients with IRSp53 mutations, IRSp53 is an important component of excitatory synapses. If IRSp53 KO mice have social deficits similar to autism-related genes with similar physiological functions, the findings of this study may have more general significance.

9. Further discussion on how changes in activity related to abnormal social behavior, or whether abnormal social behavior cause decreased release e.g., of a neuromodulator, that cause these changes in neural activity.

*Reviewer #2 (Recommendations for the authors):*

The broader significance of these findings is questionable because the relevance of this model to autism is unclear, the actual magnitudes of differences between genotypes are very small, and the relationship of changes in neural activity to either underlying mechanisms or behavior is unclear. When you put all of these things together, it's unclear how these findings lead to a general insight into how the brain works or autism. These issues could be addressed if a) the link between this gene and autism was much stronger, b) there was some exploration of the effects of NMDAR antagonists which reverse behavioral deficits such as memantine, and/or c) there was substantially greater exploration of how bursting contributes to encoding social information that lead to new insights into fundamental biology. As it stands, while I find the results interesting, it's hard to see what is the big insight that would justify publication in *eLife* vs. a more specialized neuroscience journal.

Additional comments:

1. There is a lot of discussion about 'decoding' but the authors do not actually examine this, e.g., by building a decoder, or even looking at single cell metrics, e.g., area under the receiver operator curve, mutual information, etc. This is critical because the dynamic range of a neuron is not the only determinant of how much information is transmitted about a stimulus. In particular, it is possible that decreased variability, either at the single cell or population level, compensates for changes in dynamic range (which are not accounted for in the discrimination index).

2. To what extent are changes in the max firing rate during social task / the discrimination index explained by changes in bursting explain all of this? What if you get rid of bursts, e.g., eliminate bursts in the spike trains – then do changes in the discrimination index, etc. go away?

3. Some intuition for / discussion about how increased NMDAR function or other neurobiological mechanisms cause decreased bursting is warranted.

4. As the authors allude to, it can be difficult to disentangle cause and effect in studies like this. E.g., are changes in activity a cause of abnormal social behavior, or does abnormal social behavior cause decreased release e.g., of a neuromodulator, that cause these changes in neural activity. This is very difficult to work out, and I don't think it is a fatal issue but some additional discussion / acknowledgement of this is warranted.

5. The authors might want to assess the proportion of social and object neurons in shuffled data to rule out the possibility that changes in the proportions of these neurons simply reflect nonspecific changes in activity. This does not seem to be the case because the proportion goes up for the empty chamber but it is easy enough to do this.

[Editors’ note: further revisions were suggested prior to acceptance, as described below.]

Thank you for resubmitting your work entitled "Suppressed prefrontal neuronal firing variability and impaired social representation in IRSp53-mutant mice" for further consideration by *eLife*. Your revised article has been evaluated by Catherine Dulac (Senior Editor) and a Reviewing Editor.

The manuscript has now been seen by all three previous reviewers. Reviewers 2 and 3 are satisfied with the revisions, while reviewer 1 suggested some additional analyses and text revisions.

*Reviewer #1 (Recommendations for the authors):*

I appreciate the significant effort the authors have put into addressing and analyzing the original concerns with this study. While the manuscript has been improved, some of my original concerns have not been fully addressed.

My original comment related to a decreased proportion of social neurons in the mPFC of IRSp53 KO mice (original Figure 7, now Figure 3) has not been appropriately addressed. I think using different z-score cut-offs to define social and object neurons is artificial. In Figures 3C and 3D, target shuffled data compared between WT and KO mice were significantly different, suggesting the comparison between non-linear fits could be oversensitive in revealing statistical significance that is unreal. Based on the data, I would suggest the authors perform an auROC analysis to define neurons responsive to either social or object cues (see Li et al., 2017 Cell from Catherine Dulac's lab).

The manuscript remains difficult to read and the rationale behind different analyses could be more clear. Some main figures describing similar changes could be combined while some non-essential results could be moved to supplementary.

*Reviewer #2 (Recommendations for the authors):*

The authors have responded to my comments in several ways including:

- Finding that social, but not nonsocial decoding is impaired in KO mice

- Showing how bursts vs. tonic firing contribute to altered firing and encoding related to social exploration

- Showing that the NMDA receptor antagonist memantine has opposing effects on burst firing in WT vs. KO mice

- Identifying social and object neurons based on shuffled data

Together, these additions address the major issues/suggestions I raised in my previous review.

*Reviewer #3 (Recommendations for the authors):*

I think the authors have done a good job responding to the earlier round of critiques. The new analyses are nice and directly address my concerns, especially on social behavior, with DLC, and on neural activity similarity across first and second S-O sessions.

---

## [Author Response]

Essential revisions:1. More evidence of social impairments in IRSp53 KO mice. Based on the results of Figure 1, the authors stated that KO animals 'display social impairments', but this may not be the only interpretation of their data. In Figure 1E, time spent sniffing the objects during the first S-O session is not significantly different between wild-types and KOs, although variability is higher in KOs; similarly, social sniff times are also variable. Perhaps the KO animals have somewhat stronger sniffs, or are faster at processing the olfactory information, requiring them to spend less time sniffing and remaining in the in-zone. Please address this comment by providing further data demonstrating social impairments in KO mice.

We appreciate this comment and attempted to take a closer look at the genotypic differences in sniffing behaviors. We divided the social interactions into two groups (proximal and distal), in which proximal interaction is defined as bi-directional social interaction in which the distance between the noses of the subject mice and social target mice is less than 2.5 cm while distal interaction is defined as social interaction occurring in the range of nose distance between 2.5 and 10 cm (Figure 1G). We found, in the first S-O session, that KO mice showed decreased social interactions in both proximal and distal zones, as compared with WT mice (Figure 1G). Intriguingly, KO mice showed significantly increased object interaction in the distal but not proximal zone (Figure 1I). In the second S-O session, KO mice showed social and object interactions comparable to those in WT mice (Figure 1H,J). The ratio of time spent in the proximal versus distal zone during target sniffing was not different between genotype (Figure 1—figure supplement 3A,B). These results collectively suggest that KO mice show suppressed social interaction in both proximal and distal zones and enhanced object interaction in the distal but not proximal zone, further suggesting impaired social interaction in KO mice.

In addition, we have already shown that the mean sniffing durations per single sniffing event are comparable between genotypes for empty, social, and object targets (Figure 1—figure supplement 2E). Instead, the number of visits to the social targets is significantly lower in the KO mice compared to the WT mice (Figure 1—figure supplement 2D). If KO animals have somewhat stronger sniffs, or are faster at processing olfactory information, the duration per sniffing event is highly likely to be shorter in the KO mice, which is not the case in IRSp53 KO mice for both social and object targets.

Other lines of evidence supporting social deficits in IRSp53-KO mice used in the present study have been previously reported, which include the results from the three-chamber test, reciprocal dyadic social-interaction test, and ultrasonic vocalization test (Chung et al., 2015). In addition, IRSp53-KO mice have been demonstrated to have normal olfactory function, as supported by the buried food seeking test (Chung *et al.*, 2015). We commented on these results in the Results section of the revised manuscript.

2. Stronger evidence that the observed changes in firing rate really reflect differences in response to social and non-social targets in WT and IRSp53 KO mice. In the identified single-units in the mPFC of either WT or IRSp53 KO, only 10% were responsive to social and/or object targets, while 90% of the recorded units are not responsive to either target (Figure 7E). Consistently, the mean firing rate in E-E, fS-O and sS-O are similar in both WT and KO mice (Figure 3-supplement 1E), indicating that the presence of social targets had little effect in regulating most of the recorded mPFC neurons. Therefore, it is not convincing that the recorded mPFC neurons play an essential role in discriminating social targets. Please examine whether the FR range and burst firing proportion are different between WT and KO in resting and non-social conditions.

We appreciate this comment as well. We performed additional analysis and now clearly show that although the firing rate range is comparable between genotypes during the resting condition, it is significantly lower in KO neurons in both E-E session (non-social condition) and S-O sessions (social conditions) (Figure 6— figure supplement 1D–G). We additionally show that the firing rate range increases during the S-O sessions compared to that during the E-E session only in WT neurons but not in KO neurons (Figure 6— figure supplement 1C). These results suggest that although reduced firing rate range in the KO mice is not absolutely specific to social conditions, its reduction is most salient in the social context.

We also show, by additional analyses, that, similar to the firing rate range, decreases in burst are present in both E-E and S-O sessions, but not in the resting session (Figure 7D, Figure 7— figure supplement 1A–C). This result suggests that the reduced burst firing is not specific to social conditions.

However, there are other possible interpretations of these results as well. Because neural activity was recorded repeatedly over multiple days, subject mice in the E-E session may remember the interaction with another social target in the past S-O sessions, and may expect a social target in advance. Therefore, E-E session may have some aspects of social context as well, which may have appeared in the form of decreased firing-rate range and burst in the KO neurons in the E-E session. In fact, we have added several new findings that suggest the possibility that burst firing may underlie impaired cortical encoding of social information in IRSp53-KO mice, which we’ll discuss in our responses to comment #4 and #7.

We also attempted to address the general concern for the need of stronger evidence for the genotype differences in firing rate using the support vector machine classifier, which is discussed in our responses to comment #4 in detail.

3. Suggestions for further data analysis. Perhaps moving the classification in Figure 7 to earlier in the manuscript, and analyzing firing rate statistics separately for 'social' vs 'non-social' neurons. Does unit classification (i.e., of 'social' or not) hold up across sessions or episodes of engagement within a session? If these really are 'social' units, presumably that aspect should be reliable across interactions with different mice.

In response, we moved the classification of social and object responsive neurons (previously Figure 7) to Figure 3.

We conducted an additional analysis on the consistency of firing in response to targets. In order to assess whether the unit classifications hold up across sessions, we plotted the mean firing rate responses (z-score) of neurons during target sniffing in the first S-O session against those in the second S-O session. We found positive correlations in both WT and KO neurons for both social and object targets (Figure 3— figure supplement 2A,B), which suggests that the responses of neurons to a target are largely consistent across sessions.

It was previously reported that the prefrontal neurons that encode social information do not engage in all social behavior epochs but rather show “trial-to-trial stochasticity’ (Liang et al., 2018). Similarly, we calculated the overall trial consistency of social and object neurons and found that both WT and KO neurons display consistencies over 50%, which suggests that responses of social and object neurons do not occur by chance. In addition, they display comparable levels of trial-to-trial stochasticity (Figure 3— figure supplement 2G–I).

In order to record different units in each recording experiment, the implanted tetrodes were lowered by 62.5 µm after each recording. Because we used the same subject mice for both S-O sessions, we were unable to assess whether our social units are reliable across interactions with ‘different’ mice. However, we fully appreciate this comment and are eager to address this in our future work.

4. Further analysis of firing rate data. To what extent are changes in the max firing rate during social task / the discrimination index explained by changes in bursting explain all of this? If one takes out the bursts, e.g., eliminate bursts in the spike trains – do changes in the discrimination index, etc. go away? Please also assess the proportion of social and object neurons in shuffled data to rule out the possibility that changes in the proportions of these neurons is due to nonspecific changes in activity. Other suggestions for further analysis include looking at single cell metrics, e.g., area under the receiver operator curve, mutual information, etc.

To address the issue of burst, we attempted to eliminate the bursts (ISI ≤ 10 msec) in the spike trains and were able to clearly show that the decreases in spike variability, firing-rate range, and discriminability between social and object targets still remain after burst elimination (Figure 7— figure supplement 2A–G). However, we were aware that simple elimination of burst spikes, post-recording, do not eliminate the real-time effects of bursts on neighboring neurons during the recording. In fact, it has been reported that burst spikes increase the probability of post-neuronal spike more effectively than tonic spikes (Krahe and Gabbiani, 2004; Lisman, 1997). Therefore, we alternatively assessed whether burst alone is sufficient for the genotypic differences in firing rate by eliminating the tonic spikes. We found that overall spike variability and firing-rate range during linear chamber exploration were still reduced in the KO neurons after tonic spike elimination (Figure 7— figure supplement 2H–L). In addition, the extent of discriminability between social and object targets were maintained in both WT and KO neurons, and still lower in KO neurons compared to WT neurons, after tonic spike elimination (Figure 7— figure supplement 2N). Interestingly, the left versus right side discrimination in the E-E session was intact in WT neurons with burst spikes alone but not in KO neurons, suggesting that burst spikes may also play a role in side discriminability (Figure 7— figure supplement 2M). While decreased burst may affect left versus right side discriminability in KO neurons, tonic spikes may be sufficient for overcoming this weakness. On the other hand, tonic spikes may not be sufficient to overcome the weakened social versus object discriminability in KO neurons. Further issue on burst is additionally addressed in comment #7.

We fully appreciate and agree that it is essential to eliminate the possibility that the decreased proportion of KO social neurons is due to non-specific factors, such as changes in target sniffing behavior, rather than changes in social target-dependent firing. To address this issue, we performed the same analysis after randomly shuffling the empty, social and object sniffing neural data, which would eliminate the factor of target specificity, only leaving variations in sniffing behavior intact. We found that the shuffling lowers the social and object neuron proportions in both WT and KO mPFC, indicating that both target specificity and behavioral factors contribute to target-dependent neural activity (Figure 3C and D, Figure 3—figure supplement 1B). To eliminate the effect of behavioral factors, we subtracted the target-shuffled data from the original neuron population. We found that social neuron proportion was still reduced in KO mPFC compared to WT mPFC even after this correction (Figure 3E). Although the genotypic difference in object neuron proportion was not as salient as the social neuron proportion after the correction, we observed a slight increase in KO population at lower z-score thresholds (Figure 3F). In addition, empty neuron proportion was generally increased in KO mPFC (Figure 3— figure supplement 1C). An increase in neuronal population encoding non-social target information was observed in Shank2-KO mice, an animal model of autism also displaying social deficit (Lee et al., 2021).

We appreciate this essential comment that we should also take into consideration neural variability as a determinant of how much information is transmitted. We performed both single cell and ensemble decoding analyses using the support vector machine to further assess the ability of neurons to discriminate between social and object targets. We found that while decoding accuracy was comparable between genotypes for left versus right side discriminability in the E-E session (Figure 5F and H), decoding accuracy was significantly lower in KO neurons compared to WT neurons for social versus object target discriminability (Figure 5G and I, Figure 5— figure supplement 1C and D).

5. Cleaner analysis relating the unit spike trains to moment-to-moment features of social interaction or other behaviors occurring during the S-O sessions. The authors use machine learning to classify mouse part position, but only seem to mark where the mouse is in the track. The unit activity over time might be much more interesting if correlated with DeepLabCut-based analysis of what the animal was doing (or perhaps what the social interaction partner was doing) during the sessions. Please correlate behavior to activity, both of individual units and of simultaneously-recorded populations.

In response, we have newly performed DeepLabCut analysis, additionally labeling the nose and body center of the social target mice (Figure 1B). In the attempt to examine the sniffing behavior in a detailed manner, we calculated the distance between the noses of subject and social target mice (Figure 1G). We noticed that there were two initial peaks in the distribution of nose-distance duration: a first peak at nose-to-nose distance ranging from 0 to 2.5 cm, defined as proximal interaction, and a second peak at nose-to-nose distance ranging from 2.5 to 10 cm, defined as distal interaction. As already mentioned in the response to comment #1, we found, in the first S-O session, that KO mice showed decreased social interactions in both proximal and distal zones, as compared with WT mice (Figure 1G). Intriguingly, KO mice showed significantly increased object interaction in the distal but not proximal zone (Figure 1I). In the second S-O session, KO mice showed social and object interactions comparable to those in WT mice (Figure 1H,J). The ratio of time spent in the proximal or distal zone was not different between genotype (Figure 1—figure supplement 3A,B). These results collectively suggest that KO mice show suppressed social interaction in both proximal and distal zones and enhanced object interaction in the distal but not proximal zone, further suggesting impaired social interaction in KO mice.

We also found that the firing rate responses during proximal social and non-social target interactions were significantly greater than distal target interactions in both genotypes (Figure 3— figure supplement 2E and F). In addition, by examining the effect of proximity for social versus non-social target discrimination, we found that although the discriminability was generally greater during proximal interaction compared to distal interaction, the discriminability in KO neurons was reduced in both proximal and distal conditions compared to that in WT neurons in first S-O session. It should be noted, however, that the discriminability was no longer significantly different between genotypes during the distal social interaction in the second S-O session (Figure 5—figure supplement 2A–D).

6. The authors found that the average resting firing rates of excitatory mPFC neurons in awake IRSp53-KO mice are larger than that in WT mice (Figure 1C and 1D). These results are opposite to their previous findings obtained from anaesthetized mice (Chung et al., 2015, Figure 8b and 8c). However, no such difference was observed between awake and anaesthetized WT animals. Please discuss.

Brain circuits would be more active during the awake state, compared to an anesthetized state (Albrecht and Davidowa, 1989; Shumkova et al., 2021). In addition, urethane has been shown to have diverse influences on neurotransmitter receptors (GABA, glycine, acetylcholine, NMDA, and AMPA receptors), synapses, and neuronal excitability (Accorsi-Mendonca et al., 2007; Hara and Harris, 2002; Sceniak and Maciver, 2006). Because of this broad activity of urethane, it would be difficult to narrow down and specify the key mechanism underlying the difference between the awake and anesthetized states in IRSp53-KO mice. In addition, we found, by additional slice electrophysiology experiments, that prefrontal layer 5 pyramidal neurons in IRSp53-KO mice show altered baseline properties and memantine (NMDAR antagonist)-induced changes in intrinsic excitability and burst firing (Figures 8A–G, Figure 8—supplement figure 1A–H), which may contribute to the altered responses of IRSp53-KO neurons to urethane.

7. Additional data on the contribution of NMDA receptors to observed changes in firing, Additional data on the effects of NMDAR antagonists could significantly strengthen the manuscript by potentially providing some mechanistic information and strengthening the correlation between changes in these physiological measures and changes in behavior. Please also provide some discussion about how increased NMDAR function or other neurobiological mechanisms cause decreased bursting.

We appreciate this thoughtful comment. Although it would be best if we could test the effects of memantine treatment on the activity of mPFC neurons of WT and KO mice in the linear chamber, because tetrode single-unit recording requires repeated recordings, we expected repeated intraperitoneal injections right before the experiment to be too stressful for the mice. Instead, we attempted slice electrophysiology to find out how firing properties change upon memantine treatment by measuring the intrinsic excitability-related parameters in layer 5 pyramidal neurons in the prelimbic region of the mPFC in WT and IRSp53-KO mice (3–6 months) in the presence and absence of memantine treatment. We chose layer 5 pyramidal neurons because deep cortical layers have been implicated in cognitive and social functions (Brumback et al., 2018; Kim et al., 2020; Murugan et al., 2017; Paulsen et al., 2022; Peixoto et al., 2016; Phillips et al., 2019; Rapanelli et al., 2019; Willsey et al., 2013; Yamamuro et al., 2020; Yan and Rein, 2021).

There were no baseline genotype differences in the parameters associated with intrinsic excitability, including current-firing curve, action potential (AP) threshold, input resistance, AP amplitude, full width at half maximum (FWHM), and afterhyperpolarization (AHP) (Figure 8A-F).

Upon memantine treatment, WT or IRSp53-KO neurons did not display strongly altered current-firing curve or AP-related parameters (Figure 8B,C; Figure 8—figure supplement 1A–F). However, memantine treatment induced increases in FWHM in both WT and IRSp53-KO neurons and in AP amplitude in IRSp53-KO but not WT neurons (Figure 8—figure supplement 1D,E), which may contribute to the increasing tendency of current-firing curve in WT and IRSp53-KO neurons (Figure 8B,C).

Importantly, when burst firing was analyzed by interspike interval (ISI) analysis of all spikes, memantine treatment decreased burst firing in WT neurons whereas it increased burst firing in IRSp53-KO neurons (Figure 8D–G).

These results collectively suggest that memantine treatment does not affect total firing in WT or IRSp53-KO neurons whereas it induces opposite changes in burst firing. Therefore, memantine seems to shift the balance towards tonic (non-burst) firing in WT neurons, but towards burst firing in IRSp53-KO neurons. In addition, these results suggest that decreased firing variability and burst firing may connect NMDAR dysfunction with social deficits in IRSp53-KO mice, and, more broadly, suggest that firing variability and burst firing may be a key modulator of prefrontal social encoding and regulation.

However, it remains unclear how IRSp53-KO prelimbic neurons show impaired target-induced increase in burst firing (Figure 7) despite that their NMDAR function is increased in layer 5 prelimbic neurons (Noh et al., 2022), and how acute memantine treatment increases and decreases burst firing in WT and IRSp53-KO neurons, respectively (Figure 8F,G). We made related discussion points in the revised Discussion section.

8. Further discussion on social behavior deficits in IRSp53 KO mice and how components of excitatory synapses contribute to autism. While there may be few autism patients with IRSp53 mutations, IRSp53 is an important component of excitatory synapses. If IRSp53 KO mice have social deficits similar to autism-related genes with similar physiological functions, the findings of this study may have more general significance.

We appreciate these comments. In the revised Discussion, we discussed how deletion of IRSp53, a component of excitatory synapses, may lead to social deficits with a focus on the impacts of IRSp53 deletion on excitatory synaptic actin dynamics, synaptic excitation/inhibition imbalance, NMDAR modulation (synaptic vs. extrasynaptic NMDARs), and firing mode (burst vs. tonic) in prefrontal neurons. These results at different mechanistic levels may have general significance on the etiology of ASD. We admit that IRSp53 is not a strong ASD-risk gene, but we updated additional clinical information on IRSp53-related neurodevelopmental disorders in the introduction of the revised manuscript.

9. Further discussion on how changes in activity related to abnormal social behavior, or whether abnormal social behavior cause decreased release e.g., of a neuromodulator, that cause these changes in neural activity.

We appreciate this comment and agree with the impacts of the changes in neural activity (i.e., discriminability and burst firing) on social behaviors, and the feedback influences of social behaviors on neural activity and related neuromodulators. We discussed these possibilities in the revised discussion.

References

Accorsi-Mendonca, D., Leao, R.M., Aguiar, J.F., Varanda, W.A., and Machado, B.H. (2007). Urethane inhibits the GABAergic neurotransmission in the nucleus of the solitary tract of rat brain stem slices. Am J Physiol Regul Integr Comp Physiol *292*, R396-402. 10.1152/ajpregu.00776.2005.

Albrecht, D., and Davidowa, H. (1989). Action of urethane on dorsal lateral geniculate neurons. Brain Res Bull *22*, 923-927. 10.1016/0361-9230(89)90001-4.

Brumback, A.C., Ellwood, I.T., Kjaerby, C., Iafrati, J., Robinson, S., Lee, A.T., Patel, T., Nagaraj, S., Davatolhagh, F., and Sohal, V.S. (2018). Identifying specific prefrontal neurons that contribute to autism-associated abnormalities in physiology and social behavior. Mol Psychiatry *23*, 2078-2089. 10.1038/mp.2017.213.

Chung, W., Choi, S.Y., Lee, E., Park, H., Kang, J., Park, H., Choi, Y., Lee, D., Park, S.G., Kim, R., et al. (2015). Social deficits in IRSp53 mutant mice improved by NMDAR and mGluR5 suppression. Nat Neurosci. 10.1038/nn.3927.

Hara, K., and Harris, R.A. (2002). The anesthetic mechanism of urethane: the effects on neurotransmitter-gated ion channels. Anesth Analg *94*, 313-318, table of contents. 10.1097/00000539-200202000-00015.

Kim, I.H., Kim, N., Kim, S., Toda, K., Catavero, C.M., Courtland, J.L., Yin, H.H., and Soderling, S.H. (2020). Dysregulation of the Synaptic Cytoskeleton in the PFC Drives Neural Circuit Pathology, Leading to Social Dysfunction. Cell Rep *32*, 107965. 10.1016/j.celrep.2020.107965.

Krahe, R., and Gabbiani, F. (2004). Burst firing in sensory systems. Nat Rev Neurosci *5*, 13-23. 10.1038/nrn1296.

Lee, E., Lee, S., Shin, J.J., Choi, W., Chung, C., Lee, S., Kim, J., Ha, S., Kim, R., Yoo, T., et al. (2021). Excitatory synapses and gap junctions cooperate to improve Pv neuronal burst firing and cortical social cognition in Shank2-mutant mice. Nat Commun *12*, 5116. 10.1038/s41467-021-25356-2.

Liang, B., Zhang, L., Barbera, G., Fang, W., Zhang, J., Chen, X., Chen, R., Li, Y., and Lin, D.T. (2018). Distinct and Dynamic ON and OFF Neural Ensembles in the Prefrontal Cortex Code Social Exploration. Neuron *100*, 700-714 e709. 10.1016/j.neuron.2018.08.043.

Lisman, J.E. (1997). Bursts as a unit of neural information: making unreliable synapses reliable. Trends Neurosci *20*, 38-43. 10.1016/S0166-2236(96)10070-9.

Murugan, M., Jang, H.J., Park, M., Miller, E.M., Cox, J., Taliaferro, J.P., Parker, N.F., Bhave, V., Hur, H., Liang, Y., et al. (2017). Combined Social and Spatial Coding in a Descending Projection from the Prefrontal Cortex. Cell *171*, 1663-1677 e1616. 10.1016/j.cell.2017.11.002.

Noh, Y.W., Yoo, C., Kang, J., Lee, S., Kim, Y., Yang, E., Kim, H., and Kim, E. (2022). Adult re-expression of IRSp53 rescues NMDA receptor function and social behavior in IRSp53-mutant mice. Commun Biol, in press.

Paulsen, B., Velasco, S., Kedaigle, A.J., Pigoni, M., Quadrato, G., Deo, A.J., Adiconis, X., Uzquiano, A., Sartore, R., Yang, S.M., et al. (2022). Autism genes converge on asynchronous development of shared neuron classes. Nature *602*, 268-273. 10.1038/s41586-021-04358-6.

Peixoto, R.T., Wang, W., Croney, D.M., Kozorovitskiy, Y., and Sabatini, B.L. (2016). Early hyperactivity and precocious maturation of corticostriatal circuits in Shank3B(-/-) mice. Nat Neurosci *19*, 716-724. 10.1038/nn.4260.

Phillips, M.L., Robinson, H.A., and Pozzo-Miller, L. (2019). Ventral hippocampal projections to the medial prefrontal cortex regulate social memory. *eLife 8*. 10.7554/*eLife*.44182.

Rapanelli, M., Tan, T., Wang, W., Wang, X., Wang, Z.J., Zhong, P., Frick, L., Qin, L., Ma, K., Qu, J., and Yan, Z. (2019). Behavioral, circuitry, and molecular aberrations by region-specific deficiency of the high-risk autism gene Cul3. Mol Psychiatry. 10.1038/s41380-019-0498-x.

Sceniak, M.P., and Maciver, M.B. (2006). Cellular actions of urethane on rat visual cortical neurons in vitro. J Neurophysiol *95*, 3865-3874. 10.1152/jn.01196.2005.

Shumkova, V., Sitdikova, V., Rechapov, I., Leukhin, A., and Minlebaev, M. (2021). Effects of urethane and isoflurane on the sensory evoked response and local blood flow in the early postnatal rat somatosensory cortex. Sci Rep *11*, 9567. 10.1038/s41598-021-88461-8.

Willsey, A.J., Sanders, S.J., Li, M., Dong, S., Tebbenkamp, A.T., Muhle, R.A., Reilly, S.K., Lin, L., Fertuzinhos, S., Miller, J.A., et al. (2013). Coexpression networks implicate human midfetal deep cortical projection neurons in the pathogenesis of autism. Cell *155*, 997-1007. 10.1016/j.cell.2013.10.020.

Yamamuro, K., Bicks, L.K., Leventhal, M.B., Kato, D., Im, S., Flanigan, M.E., Garkun, Y., Norman, K.J., Caro, K., Sadahiro, M., et al. (2020). A prefrontal-paraventricular thalamus circuit requires juvenile social experience to regulate adult sociability in mice. Nat Neurosci *23*, 1240-1252. 10.1038/s41593-020-0695-6.

Yan, Z., and Rein, B. (2021). Mechanisms of synaptic transmission dysregulation in the prefrontal cortex: pathophysiological implications. Mol Psychiatry. 10.1038/s41380-021-01092-3.

[Editors’ note: further revisions were suggested prior to acceptance, as described below.]

Reviewer #1 (Recommendations for the authors):I appreciate the significant effort the authors have put into addressing and analyzing the original concerns with this study. While the manuscript has been improved, some of my original concerns have not been fully addressed.

We appreciate the reviewer for acknowledging the improvements made in our revised manuscript. We made additional changes to address the remaining concerns of the reviewer, as described below.

My original comment related to a decreased proportion of social neurons in the mPFC of IRSp53 KO mice (original Figure 7, now Figure 3) has not been appropriately addressed. I think using different z-score cut-offs to define social and object neurons is artificial. In Figures 3C and 3D, target shuffled data compared between WT and KO mice were significantly different, suggesting the comparison between non-linear fits could be oversensitive in revealing statistical significance that is unreal. Based on the data, I would suggest the authors perform an auROC analysis to define neurons responsive to either social or object cues (see Li et al., 2017 Cell from Catherine Dulac's lab).

Following this suggestion, we re-analyzed the proportion of neurons responsive to social and object targets using an auROC analysis (referring to Li et al., 2017 Cell paper). Because the results of the current analysis matched those of the z-score analysis, we decided to replace the z-score results with the auROC results (Figure 3). We believe that the current results are clearer than before, thanks to the comment of the reviewer.

The manuscript remains difficult to read and the rationale behind different analyses could be more clear. Some main figures describing similar changes could be combined while some non-essential results could be moved to supplementary.

We also appreciate the comment that the manuscript is difficult to read. As suggested by reviewer #1, we reduced the number of main figures from 8 to 7, and moved some non-essential results to supplementary figures. Moreover, some results that we found redundant were deleted from the manuscript.

– We moved the results in Figure 4 to a new supplementary figure (Figure 5—figure supplementary 2).

– We deleted the results of the discrimination index analysis (Figure 5D,E and Figure 5—figure supplement 1B) from our manuscript because it is redundant with the results of the SVM analysis. We moved the single cell SVM results in Figure 5F,G to a new supplementary figure (Figure 4—figure supplementary 1B,C).

– We deleted the results from the ISI coefficient of variance (CV) analysis (Figure 7F–H), because it is redundant with the firing rate variability (σ) results. We also moved the results in figure 7D,E to a new supplementary figure (Figure 6—figure supplementary 1A,B).